# Global impacts of aviation on air quality evaluated at high resolution

Sebastian D. Eastham*[1,2], Guillaume P Chossière[1], Raymond L. Speth[1,2], Daniel J. Jacob[3], Steven R.H. Barrett[1,2]

[1]Laboratory for Aviation and the Environment, Department of Aeronautics and Astronautics, Massachusetts Institute of Technology, Cambridge, MA 02139, USA
[2]Joint Program on the Science and Policy of Global Change, Massachusetts Institute of Technology, Cambridge, MA 02139, USA
[3]Atmospheric Chemistry Modeling Group, John A. Paulson School of Engineering and Applied Sciences, Harvard University, Cambridge, MA 02138, USA

*Correspondence to*: Sebastian D. Eastham (seastham@mit.edu)

**Abstract.** Aviation emissions cause global changes in air quality which have been estimated to result in ~58,000 premature mortalities per year, but this number varies by an order of magnitude between studies. The causes of this uncertainty include differences in the assessment of ozone exposure impacts and in how air quality changes are simulated, and the possibility that low-resolution (~400 km) global models may overestimate impacts compared to finer-resolution (~50 km) regional models. We use the GEOS-Chem High Performance chemistry-transport model at a 50 km global resolution, an order of magnitude finer than recent assessments of the same scope, to quantify the air quality impacts of aviation with a single internally consistent, global approach. We find that aviation emissions in 2015 resulted in 21,200 (95% confidence interval due to health response uncertainty: 19,400 – 22,900) premature mortalities due to particulate matter exposure and 53,100 (36,000 – 69,900) due to ozone exposure. Compared to a prior estimate of 6,800 ozone-related premature mortalities for 2006 our central estimate is increased by 5.6 times due to the use of updated epidemiological data which includes the effects of ozone exposure during winter, and by 1.3 times due to increased aviation fuel burn. The use of fine (50 km) resolution increases the estimated impacts on both ozone and particulate matter-related mortality by a further 20% compared to coarse-resolution (400 km) global simulation, but an intermediate resolution (100 km) is sufficient to capture 98% of impacts. This is in part due to the role of aviation-attributable ozone, which is long-lived enough to mix through the Northern Hemisphere and exposure to which causes 2.5 times as much health impact as aviation-attributable $PM_{2.5}$. This work shows that the air quality impacts of civil aviation emissions are dominated by the hemisphere-scale response of tropospheric ozone to aviation $NO_x$ rather than local changes, and that simulations at ~100 km resolution provide similar results to those at two times finer spatial scale. However the overall quantification of health impacts is sensitive to assumptions regarding the response of human health to exposure, and additional research is needed to reduce uncertainty in the physical response of the atmosphere to aviation emissions.

# 1    Introduction

Aviation is a unique source of climate and air quality impacts. For example, the contrails (ice clouds) which form in aircraft exhaust do not occur for any other major mode of transportation, but have been estimated to cause as much climate forcing as all of the carbon dioxide emitted during flight (Lee et al., 2020). Similarly, the nitrogen oxides ($NO_x$) and other species emitted during flight can have long-lasting chemical consequences which result in global-scale degradation of air quality (Eastham and Barrett, 2016). It has been estimated that emissions during the cruise portion of the flight specifically contribute around 80% of the 8,000 – 58,000 premature mortalities each year attributable to aviation emissions (Barrett et al., 2010a; Eastham and Barrett, 2016; Quadros et al., 2020). When monetized, these air quality impacts are similar in magnitude to the net climate costs of aviation, including $CO_2$ and contrails (Grobler et al., 2019).

However, there remain several key uncertainties regarding the impacts of aviation emissions on air quality due to the practical challenges associated with simulating a global influence on a local quantity. Global models are well suited to quantify the global change in oxidative capacity due to aviation emissions, but must simulate the entire atmosphere to do so. Models used in previous studies have split the atmosphere up into grid cells which are between 2 and 5° (latitude and longitude) along each side, or roughly 200 – 500 km (Barrett et al., 2010a; Eastham and Barrett, 2016). This means that the models artificially diffuse local, airport-scale emissions over the surrounding area, potentially failing to resolve the co-location of near-airport emissions and exposed populations and underestimating the relative contribution of non-cruise emissions to air quality. A study by Punger and West (2013) found that, due to this co-location effect alone, coarse resolution (> 250 km) global models would likely be biased low by 30-40% when estimating US population exposure to $PM_{2.5}$. The use of large grid cells also means that models treat areas of up to 20,000 $km^2$ over a city as being a single, well-mixed air mass, and will not be able to resolve non-linear chemical processes which could increase or decrease the air quality response to aviation emissions. If the population mean exposure to air pollution varies strongly with resolution, air quality impact estimates from coarse models could therefore provide misleading results.

Studies with nested, regional models can address this question for limited areas but incur an inconsistency at the model boundary, either in resolution or in the model being used to quantify impacts. These studies calculate global atmospheric composition changes at a relatively coarse resolution, and then use those results to provide the boundary conditions for a finer-resolution simulation. Whereas some studies using nested regional models have found air quality impacts from cruise emissions of a similar magnitude to those from global studies (Yim et al., 2015; Quadros et al., 2020), Vennam et al. (2017) found that the finer-resolution nested model produced changes in surface ozone and fine particulate matter ($PM_{2.5}$) which were 70 and 13 times smaller respectively than in a global model. This finding has been used to argue that aviation's effects on air quality have been overestimated due to coarse-resolution models being unable to resolve local-scale effects (Lee et al., 2023).

A related question is whether the air quality impacts of aviation are dominated by local sources (e.g. landing and takeoff (LTO) operations and flight through local airspace) or are the result of larger atmospheric changes. If the former is true, then

regional regulations and emissions standards applied for only near-surface operations may be sufficient to reduce impacts.

However, if impacts are dominated by large scale atmospheric responses to global aviation, then emissions standards will only be effective if applied both globally and to fuel burn beyond LTO – since LTO accounts for only 9.1% of the global total from aviation (Simone et al., 2013).

These questions urgently need to be resolved. New regulations for aviation $NO_x$ emissions are being debated by the International Civil Aviation Organization. Since the establishment of the 1981 CAEE standard limiting $NO_x$ emissions per

unit of thrust for landing and take-off operations (LTO), subsequent CAEP regulations have continued to increase stringency, resulting in the current CAEP/8 standard which was set in 2010 (ICAO, 2017). Since the amount of $NO_x$ produced during take-off and during cruise are closely related for most current combustor architectures, these regulations are also the relevant limiting factor for cruise $NO_x$ emissions. Recent studies have suggested that the climate impacts of greater $NO_x$ emissions are sufficiently small relative to the benefits of reduced $CO_2$ that fuel efficiency should be prioritized over

further $NO_x$ emissions reduction (Skowron et al., 2021). However, if air quality impacts are included in this analysis then such a prioritization could cause net environmental damages rather than improvement – dependent on accurate estimation of the air quality consequences (Miller et al., 2022).

This study quantifies the global air quality response to aviation in a single, consistent modeling framework, evaluating both the role of model grid resolution and the relative contribution of local and remote aviation emissions. We use three different

global resolutions to quantify how grid resolution affects the simulated outcomes, varying from 400 km to 50 km globally. To isolate the role that in-domain ("local") and out-of-domain ("remote") emissions have on air quality in the contiguous United States we use a perturbation approach, performing an additional simulation at each resolution in which aviation emissions over the United States are set to zero. These are supplemented by sensitivity simulations described in Section 2.3. All calculations are performed with the objective of quantifying potential benefits of rapidly eliminating aviation emissions,

whether through policy or technological approaches such as post-combustion emissions control (Prashanth et al., 2020).

## 2    Method

We simulate aviation's impacts on global air quality at three resolutions, ~400 km, ~100 km, and ~50 km, without the use of regional refinement or nesting. For each of the three resolutions, we perform three simulations. We first simulate global atmospheric composition using version 12.6.2 of the GEOS-Chem High Performance model (GCHP) (Eastham et al., 2018)

with all aviation emissions enabled (AVGLOBAL) and with all aviation emissions disabled (AVOFF). The simulated differences in concentrations of ozone and particulate matter ($PM_{2.5}$) in the surface layer are then taken as the total air quality impacts of aviation, with health impacts calculated as discussed below.

These simulations are followed by a further simulation in which aviation emissions are included everywhere except directly on or above the contiguous United States within a domain covering 10-60°N and 60-130°W (AVNOUS). This domain has

previously been used in nested model simulations of regional air quality change, and can therefore be considered to be

representative of models which are focused on regional change only (Kim et al., 2015; Hu et al., 2018). The difference in air quality changes between the AVGLOBAL and AVNOUS is then taken as the effect of in-region emissions on regional air quality. The remaining differences in surface air quality within the region, calculated as the difference in air quality between AVNOUS and AVOFF, then correspond to the effect of out-of-region emissions on surface air quality. Figure 1 shows
global distribution of fuel burn and the region in which emissions are set to zero when estimating influence of "out-of-region" emissions on "in-region" air quality.

To quantify the effect of model resolution, we perform the above set of three simulations at three different global resolutions. The GCHP model uses a cubed-sphere model grid with finite volume advection (Putman and Lin, 2007). Model resolutions are therefore denoted as CN where N is the number of grid elements along each edge of the cube such that higher numbers
correspond to finer grid resolution. We perform our simulations at global resolutions of C180, C90, and C24, in which the average side lengths of a grid cell is 51 km, 100 km, and 380 km respectively. As such, they are also approximately equivalent to global resolutions of 0.5°×0.625°, 1°×1.25°, and 4°×5°, but without distortion in grid cell size near the poles and equator. By comparing the air quality impacts calculated at each model resolution, we quantify the effect that increasing model resolution has on simulated air quality. This approach uses a single, global model to accomplish this, avoiding
discrepancies due to the use of regional models or different chemical mechanisms (Yim et al., 2015; Vennam et al., 2017).

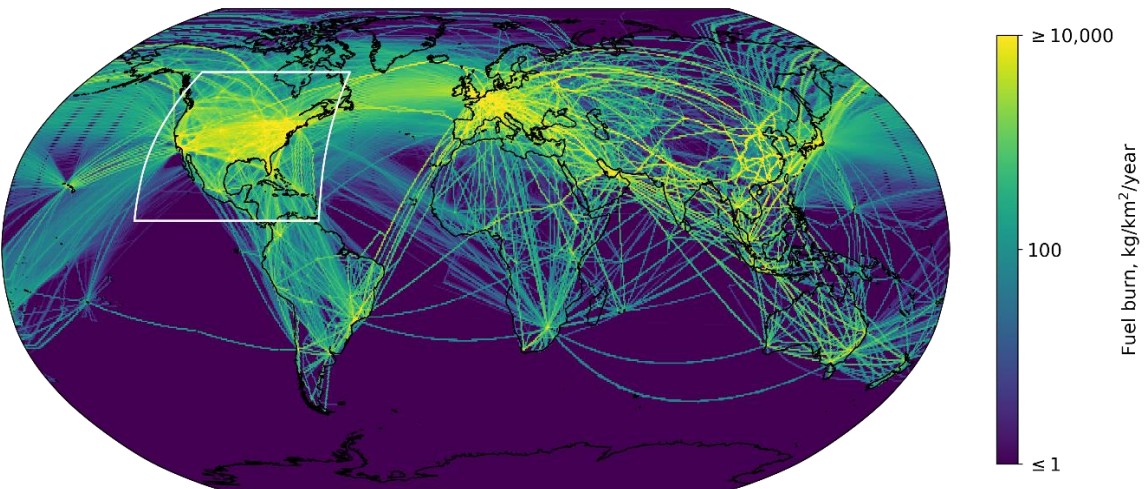

**Figure 1. Simulated global distribution of fuel burn in 2015. Data shown include fuel burned at all altitudes, at an approximate global resolution of 0.5°×0.625°. Emissions over the contiguous United States within the white box are set to zero when estimating the net effect of "out-of-region" aviation emissions on aviation's air quality impacts in the contiguous US.**

**2.1    Atmospheric simulation**

We use the global chemistry transport model (CTM) GEOS-Chem version 12.6.2 (https://doi.org/10.5281/zenodo.3543702) to simulate all scenarios, as implemented in the GEOS-Chem High-Performance model (Eastham et al, 2018). GEOS-Chem includes unified tropospheric-stratospheric chemistry and has previously been used in a global, coarse-resolution

configuration to estimate aviation's impacts on surface air quality (Eastham and Barrett, 2016; Eastham et al., 2014).

Multiple horizontal resolutions are used as described above, but all share a common vertical discretization using 72 non-uniform layers extending from the surface to a maximum altitude of around 80 km. Each simulation is integrated forwards in time for a total of 13 months from July $1^{st}$ 2014 through to July $31^{st}$ 2015, of which the final 12 are used to calculate annual mean changes in surface ozone. This period is used to ensure that the northern hemisphere winter season (December through February, inclusive) is from a single, continuous period, as this is when the greatest air quality impacts from aviation are

expected (Eastham and Barrett, 2016). All simulations use meteorological data from the NASA Global Modeling and Assimilation Office Modern Era Retrospective for Research and Analysis version 2 (MERRA-2) reanalysis product.

Emissions from civil aviation in 2015 are calculated using the Federal Aviation Administration (FAA) Aviation Environmental Design Tool (AEDT) (Wilkerson et al., 2010). This includes all emissions from civil airliners in that year during taxi, take-off, climb, cruise, descent, and landing operations. We do not account for military flights or business jets.

Emissions are gridded at a resolution of $0.25°×0.25°$ globally, finer than the highest resolution model grid (C180). A fuel sulfur content of 600 ppm by mass is assumed for all scenarios, unless otherwise stated (Hileman et al., 2010). All nitrogen oxide ($NO_x$) and volatile organic compound (VOC) emissions are speciated as described in AEDT guidance documents (Barrett et al., 2010b). Table 1 shows the total emitted mass of each relevant compound. Black and organic carbon (BC and OC) emissions are estimated using a fixed emissions factor of 30 mgC per kg of fuel burned for each species. Emissions of a

secondary organic aerosol precursor (SOAP) are calculated as 69 g of carbon per kilogram of CO emitted (Kim et al., 2015). GEOS-Chem uses a bulk aerosol parameterization and therefore does not require a size distribution or calculation of the number of particles emitted. The impacts of water vapor emissions and condensation trails on atmospheric composition are not included in this analysis.

**Table 1. Total emitted mass for the fleet in 2015. The mass basis for the reported quantity is given separately for each species. All**
**quantities are shown to two significant figures. The rightmost column shows the fraction of global emissions which occur in the simulated North American domain over the US. *US HC emissions data are not separately archived so this number is estimated.**

|  | Global | US domain | % in US domain |
|---|---|---|---|
| $NO_x$ as NO, $NO_2$, and HONO (TgN) | 1.1 | 0.17 | 15% |
| Carbon monoxide (Gg) | 590 | 130 | 22% |
| Hydrocarbons (Gg $CH_4$ mass equivalent*) | 62 | 15 | 24% |
| Soot (black carbon, GgC) | 7.2 | 1.2 | 17% |
| Organic carbon aerosol (GgC) | 7.2 | 1.2 | 17% |
| SOA precursor (GgC) | 41 | 9.2 | 22% |
| Sulfur dioxide (GgS) | 140 | 23 | 17% |
| Sulfate aerosol (GgS) | 2.9 | 0.48 | 17% |

| | Total fuel burn (Tg) | 240 | 40 | 17% |
|---|---|---|---|---|

Non-aviation emissions are provided by a collection of standard inventories, described in full in the Supplemental Information.

GEOS-Chem has been extensively used for air quality assessments in the past, including for evaluations of the effects of aviation (Quadros et al., 2020; Barrett et al., 2010a; Yim et al., 2015), long-range air pollution (Vohra et al., 2021; Huang et al., 2017), and regional-scale changes (Potts et al., 2021; Vohra et al., 2021) on air quality. A multi-model intercomparison performed by Cameron et al. (2017) found that the effects calculated by GEOS-Chem of aviation on surface air quality were consistent with estimates from other widely-used global atmospheric models including the Community Atmosphere Model 5

and the GEOS-5 Earth System Model. GEOS-Chem has also been continuously evaluated against global observations of both atmospheric composition and air quality (Zhang et al., 2011; Christian et al., 2018; Dasadhikari et al., 2019; Quadros et al., 2020). Nonetheless the results of this study reflect only the estimates from a single model.

## 2.2    Health impact estimation

Air quality impacts are calculated based on the difference in concentrations of ozone and $PM_{2.5}$ in the lowermost simulated

atmospheric layer. The lowermost layer is approximately 120 meters thick for a surface pressure of 1013.25 hPa, and the average concentration within this surface layer is treated as the exposure-relevant value. Model predictions are not-bias corrected to observations, in part because the relatively small effect of aviation emissions on ozone and particulate matter has a different spatial pattern than background concentrations (Cameron et al., 2017). Bias correction may therefore impose a non-physical pattern on the changes in surface concentration due to aviation.

Impacts are calculated on the 30 arc-second ($1/120^{th}$ of a degree) global grid on which population density data is provided by the Gridded Population of the World version 4.11 (Center for International Earth Science Information Network - CIESIN - Columbia University, 2016). The concentrations of ozone and $PM_{2.5}$ are taken from whichever simulation grid cell contains that 30 arc-second cell, with no interpolation. If the population cell straddles the boundary between two simulation grid cells, the area-weighted mean concentration is used. The age distribution and baseline mortality rates within each grid cell are

supplied by the World Health Organization (WHO) through their 2016 Global Health Estimates (World Health Organization, 2018). For non-member countries of the WHO, we use the world region mean rate instead from the same source.

In each grid cell and for each age bracket, we calculate the relative risk of mortality due to chronic exposure to ozone and $PM_{2.5}$ with and without aviation ($RR_{BASE}$ and $RR_{NOAV}$ respectively). The change in the annual mortality ($\Delta M$) due to some disease for that age bracket is then calculated for each grid cell as


$$\Delta M = M_{BASE} \times \frac{RR_{NOAV} - RR_{BASE}}{RR_{BASE}} \qquad (1)$$

where $M_{BASE}$ is the number of mortalities due to that disease in 2016. The relative risk is calculated by comparing the simulated exposure-relevant concentration without aviation ($\chi_{NOAV}$) to the concentration simulated when aviation is included ($\chi_{BASE}$) using an appropriate concentration response function. In this case $\Delta M$ is expected to be negative, implying that reducing aviation emissions to zero would reduce mortality rates. The mortality due to aviation reported in this paper is therefore $-1 \times \Delta M$. For both ozone and $PM_{2.5}$ exposure we first calculate the relevant daily quantity and then average over the year. Our evaluations focus on the link between long-term (chronic) exposure and increased mortality, rather than acute impacts associated with short-term increases in exposure.

For ozone, the increase in relative risk of mortality is calculated based on the association between exposure and mortality identified by Turner et al. (2015). Said study analyzed a cohort of 669,046 participants in the American Cancer Society Cancer Prevention Study II from 1982 to 2004, finding a 12% increase (95% confidence interval: 8.0-16%) in respiratory mortality per 10 ppb increase in annual mean, maximum daily 8-hour average (MDA8) ozone concentration. We use this data in a log-linear concentration response function (CRF), such that the relative risk due to ozone exposure is calculated as

$$RR_{NOAV} = \exp(\beta_{LL}[\chi_{NOAV} - \chi_{BASE}]) \qquad (2)$$

where the central value of $\beta_{LL}$ is calculated as $\ln(1.12)/10 = 0.011$ ppb$^{-1}$. Only adults over the age of 30 are included when calculating the increase in mortality, as this was the cohort in which the relationship was observed. Uncertainty in the concentration response function is quantified by treating $\beta_{LL}$ as a triangularly-distributed random variable. We use $\ln(1.12)/10$ as the mode and fit a triangular distribution such that $\ln(1.06)/10$ lies at 2.5% and $\ln(1.18)/10$ as 97.5% along the cumulative distribution function.

For particulate matter, relative risks are calculated using the Global Exposure Mortality Model (GEMM) (Burnett et al., 2018). The GEMM is a set of non-linear concentration response functions which estimate the increase in relative risk of mortality based on associations with the annual-average, 24-hour mean $PM_{2.5}$ mass concentration at standard temperature and pressure. We apply the age-specific GEMM CRF for combined non-communicable disease and lower respiratory infection (NCD+LRI). The parameters for the GEMM are constructed based on a meta-analysis of 41 cohort studies worldwide examining the relationship between exposure to fine particulate matter and non-accidental mortality, and are described in detail in Burnett et al. (2018).

Impacts are calculated by performing 1,000 random draws of the CRF parameters in a paired Monte-Carlo simulation. From this, the mean and 95% confidence interval of mortality due to individual sources and mortality due to ozone and $PM_{2.5}$ combined can be estimated.

## 2.3 Sensitivity simulations

The core simulations described above allow us to quantify the total health impacts which could be avoided by eliminating aviation emissions, the sensitivity of that estimates to model resolution, and the degree to which in-region emissions control in-region outcomes. In order to better understand the causes of these impacts and their limitations, we also perform a series

of additional sensitivity simulations at C90 resolution. We first repeat the core simulations (AVGLOBAL, AVNOUS, and AVOFF) using a different version (14.2.2) of the GCHP model (see Supplemental Information). This allows us to assess the degree to which changes in chemistry, background emissions, and process representation can affect the overall health impact associated with aviation. We also extract data in each of these simulations about chemical production of ozone and global chemical loss rates to support analysis of the chemistry and fate of aviation emissions (Section 3.3).

We then simulate two shorter sensitivity scenarios with the updated model in which aviation $NO_x$ is eliminated globally (AVNONOX) and aviation emissions are simulated above 1 km altitude only (AVNOLTO). These simulations allow us to establish to what extent aviation $NO_x$ (rather than other emissions) or LTO emissions (rather than those from cruise) are responsible for changes in population exposure to both ozone and $PM_{2.5}$. These simulations cover the period August 2014 to February 2015 (inclusive) in order to capture the wintertime response to aviation.

Finally, two simulations are performed at resolutions of C24 and C90 which evaluate how the rate of vertical mixing is affected by model resolution by estimating the concentration of radioactive tracer [7]Be throughout the atmosphere. This test has been previously used to assess atmospheric model accuracy in simulating vertical transport (Yu et al., 2018; Liu et al., 2001) and provides a benchmark for the degree to which high-altitude atmospheric composition can affect surface conditions. A full listing and classification of all 16 simulations is provided in the Supplemental Information (Table S1).

## 3    Results

Based on results generated at ~50 km resolution globally, we find that the increase in $PM_{2.5}$ exposure results in an additional 21,200 (95% confidence interval due to uncertainties in health response: 19,400 to 22,900) mortalities globally each year. Aviation-attributable ozone exposure results in an additional 53,100 (36,000 to 69,900) mortalities. Combined, we estimate that aviation results in 74,300 mortalities (57,300 to 91,100), or 311 mortalities per teragram of fuel burned. These results are analyzed in Sections 3.1 to 3.4 and compared to prior studies in Section 3.5.

Figure 2 shows how the changes in surface concentrations are distributed, both in the US (left) and globally (right). Changes in ozone are diffuse throughout the Northern Hemisphere, reaching a maximum over the Himalayan plateau and the Western United States. Changes in total particulate matter are more heterogeneous, with the greatest increases occurring over western China, northern India, and western Europe. This effect is likely driven by the effect of increased oxidant concentrations acting on non-aviation precursor gas emissions (Eastham and Barrett, 2016). Increases in soot (black carbon) are shown as soot is rapidly removed by precipitation, and therefore highlights near-airport regions as these are the locations most strongly affected by direct particulate matter emissions from aircraft. However, peak concentrations of aviation-attributable soot are two orders of magnitude smaller than the more diffuse increases in total $PM_{2.5}$.

Our findings suggest that direct exposure to cruise-attributable ozone is the largest air quality concern for aviation, causing 2.5 times as many premature mortalities as $PM_{2.5}$. This means that the dominant air quality impact of aviation is not localized

to airports, but is also not globally or regionally homogenous. Although aviation-attributable ozone is present throughout the Northern Hemisphere with a mean mixing ratio of 1.1 ppb, elevated regions can reach annual average mixing ratios in excess of 3 ppb. In these regions air more rapidly reaches the surface from the cruise level, meaning that less of the cruise-altitude ozone is destroyed before the air reaches the population. This is consistent with studies which have shown that surface ozone

240    in the Western US can be strongly influenced by stratospheric intrusions (Lin et al., 2012) and is analyzed in Section 3.3. Although the greatest increases are on the Himalayan plateau, aviation-attributable ozone in the western US also exceeds 2 ppb. This is due to the descent of upper tropospheric air in the lee of mountain ranges, such that the largest impacts of aviation emissions on US air quality are just east of the Rocky Mountains.

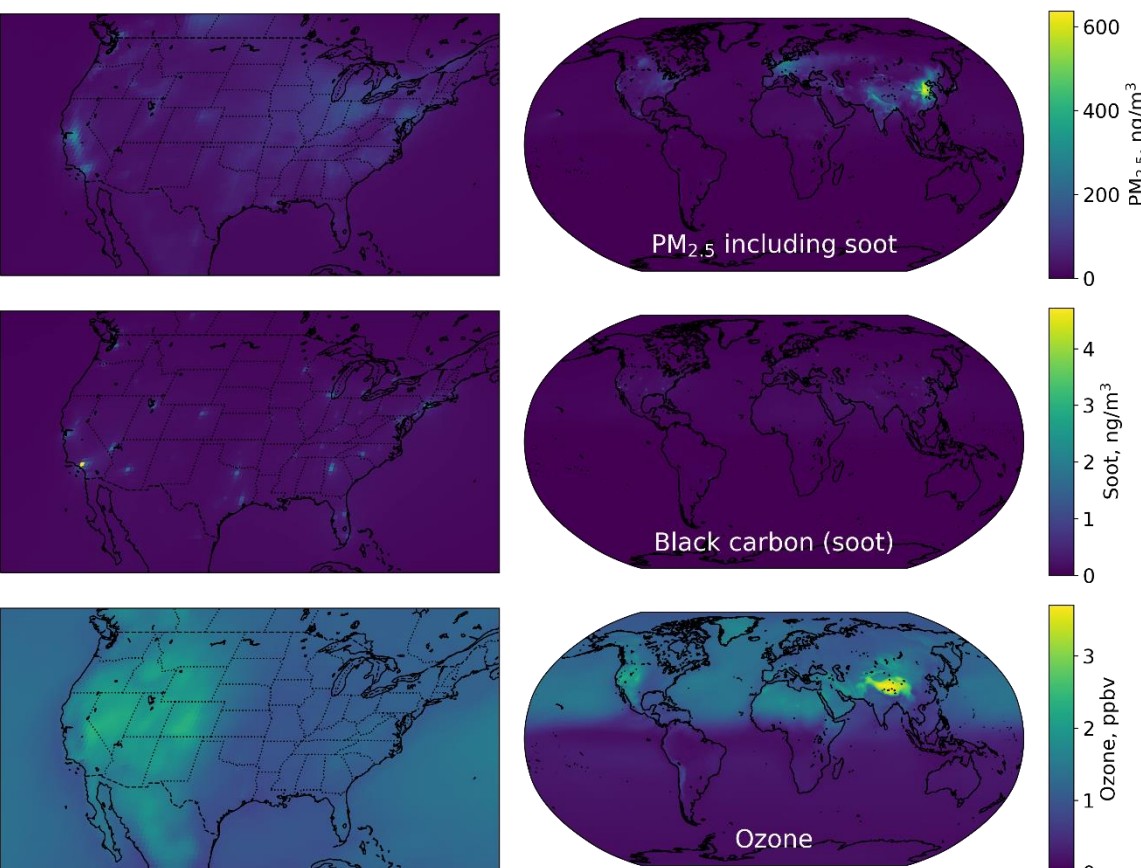

245

**Figure 2. Changes in surface air quality due to global aviation. From top to bottom: changes in PM2.5 (including soot), changes in soot only, and changes in ozone. The left column is a zoomed-in version of the right plot, focusing on the contiguous United States. Soot is explicitly singled out due to its potential as an indicator of the direct influence of aviation exhaust without any chemical intermediaries.**

250    The distribution of aviation-attributable $PM_{2.5}$ follows a different pattern. Aviation-attributable soot is concentrated near airports, but this peaks at less than 5 ng/m$^3$ (see Section 3.1). The greatest changes in $PM_{2.5}$ exceed 600 ng/m$^3$ and are spread

over larger areas in locations which already have elevated background pollution concentrations, such as western China, central Europe, and the US Northeast.

Figure 3 further explores the factors contributing to aviation-attributable increases in surface $PM_{2.5}$. The change in surface $PM_{2.5}$ due to aviation is correlated with the concentration of background (non-aviation) $PM_{2.5}$, calculated as the concentration with all other sources included, with an $R^2$ of 0.495. However, aviation-attributable $PM_{2.5}$ concentration is less strongly correlated ($R^2$ of 0.140) with the change in aviation-attributable black carbon, an indicator of nearby aviation activity. This is consistent with the hypothesis that increases in $PM_{2.5}$ due to aviation for a location in the Northern Hemisphere occur because aviation-attributable ozone contributes to the formation of secondary particulate matter from existing precursors.

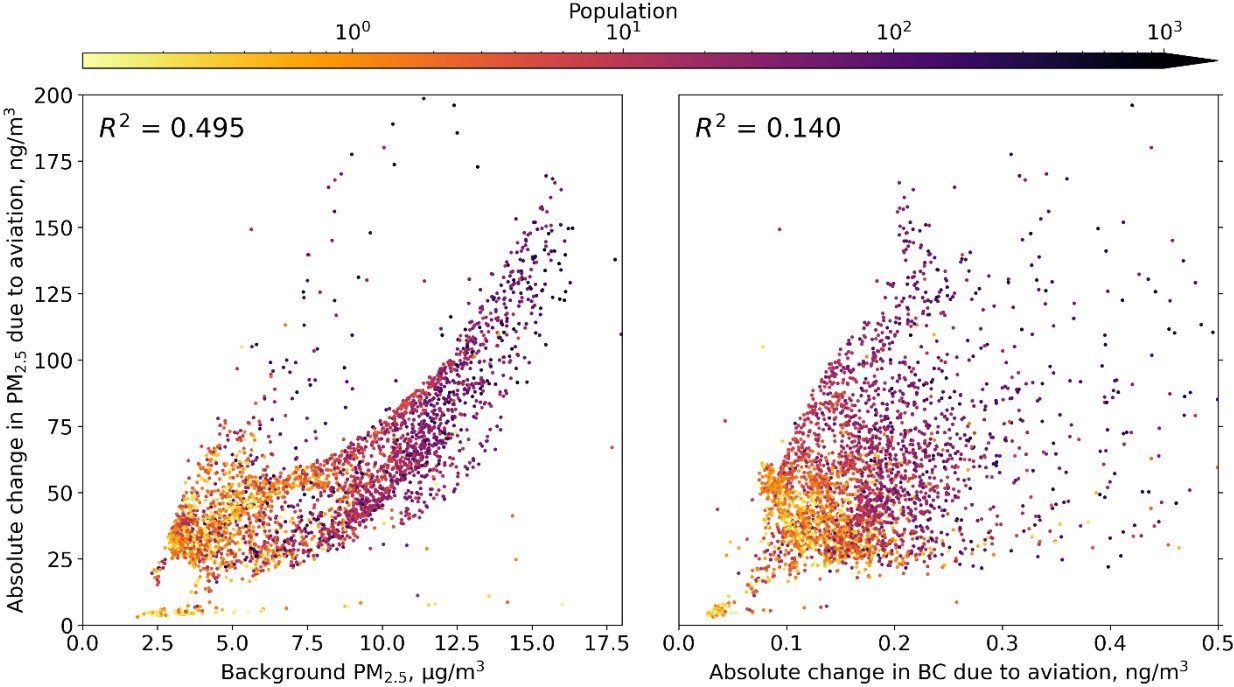

**Figure 3. Scatter plots of the change in $PM_{2.5}$ due to aviation as a function of other variables. Left: the absolute change in surface $PM_{2.5}$ due to aviation plotted as a function of the background (non-aviation) concentration of $PM_{2.5}$, colored by the total population in the grid cell. Right: aviation-attributable $PM_{2.5}$ plotted as a function of the absolute change in black carbon due to aviation. Each point corresponds to a single grid cell at C180 resolution. Only grid cells in the Northern Hemisphere with a population of at least 0.1 people per square kilometer are shown. $R^2$ values are based on a linear least-squares fit.**

### 3.1    The effect of resolution on the simulated air quality impacts of aviation emissions

Figure 4 shows the population-mean exposure to $PM_{2.5}$ (including soot), soot, and ozone for the global mean, China, the 27 European Union member states ("EU"), and the US when simulations are carried out at three different resolutions. All differences are given relative to the aviation-attributable change calculated at C180 unless otherwise stated. We find that simulations performed at the coarsest, C24 (~400 km) resolution result in a global mean ozone exposure which is 17% lower than when calculated at the C180 (~50 km) resolution, compared to 3.0% lower at C90. For $PM_{2.5}$ the exposure is 26 and

1.2% lower respectively, whereas for soot it is 18 and 6% lower. In total, the net mortalities calculated when performing simulations at a resolution of ~50 km are 2.5% higher than at ~100 km, and 25% higher than at ~400 km resolution.

The greater resolution has two effects: physical phenomena are more finely resolved, and changes in surface air quality which occur local to population centers can be more accurately collocated. These effects can be separated by performing exposure calculations using the same high-resolution (~50 km) output data but downgrading it to low resolution (~400 km) before calculating exposure. Doing so increases the calculated ozone exposure by 3.7% but decreases calculated $PM_{2.5}$ exposure by 14%. This suggests that the higher-resolution simulation of atmospheric phenomena is more important than population collocation for ozone but that the collocation effect is significant for $PM_{2.5}$, consistent with prior work focused on non-aviation exposure (Punger and West, 2013).

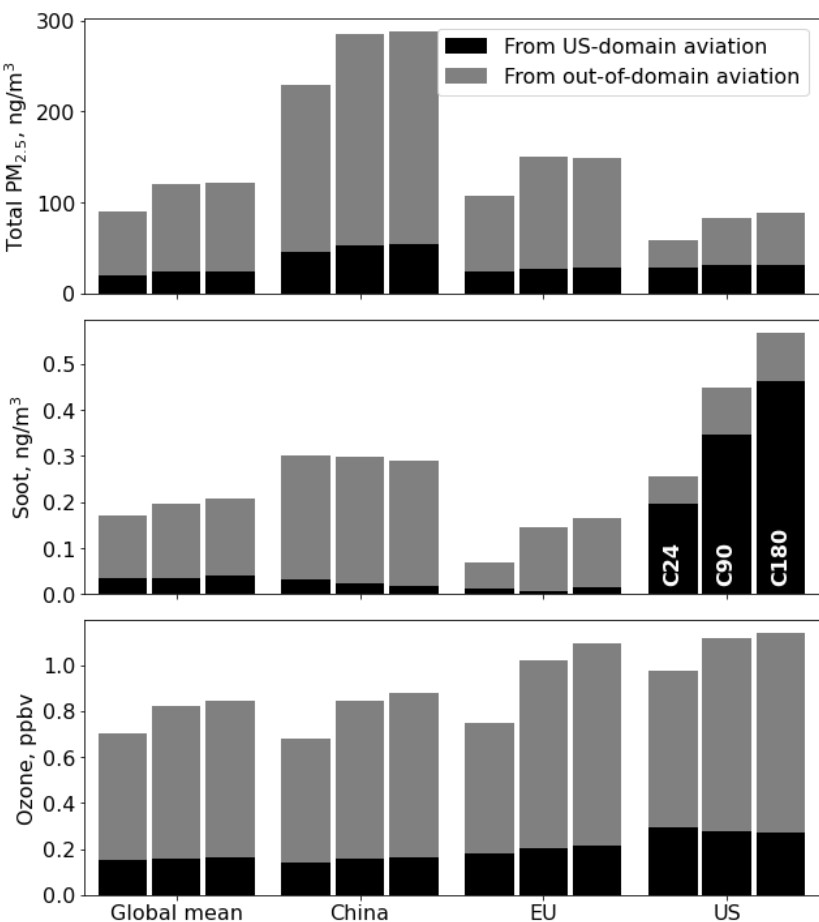

**Figure 4. Changes in air quality resulting from aviation emissions, calculated for three different resolutions (C24, C90, and C180 from left to right). Black bars show the contribution of emissions over the US only, while grey bars show the contribution of non-US domain emissions.**

This effect is further illustrated by the different simulated effect of aviation on regional and near-airport air quality at different resolutions. At both C90 and C180 resolution, the pattern of change in surface-level PM$_{2.5}$ and ozone is qualitatively similar, and the greatest changes are not necessarily in the immediate vicinity of airports. Figure S2 shows that, over the US West Coast, PM$_{2.5}$ is increased most in the California central valley and ozone is increased most over the Plains States, with the same effect visible at both C180 and C90. However, at C24 these patterns are not well resolved. Exposure to black carbon (soot) is greater in the vicinity of major airports and more accurately captured at C180, but even in a hotspot around Los Angeles International and Santa Monica airports where soot concentrations are increased by 4.7 ng/m$^3$, only 2.7% of the aviation-attributable increase in PM$_{2.5}$ is due to soot. The rest is due to a more diffuse increase in PM$_{2.5}$ extending throughout the California central valley which is captured by simulations at both C90 and C180, but which at C24 is diffused over most of California. We explore the potential causes of differences in estimated ozone and PM$_{2.5}$ changes in more detail in Section 3.3.

## 3.2    In-region versus out-of-region emissions

Concerning the role of long-range transport compared to in-region emissions, of the 21,200 mortalities due to aviation-attributable exposure to PM$_{2.5}$, 1,610 (1,470 to 1,740) occur in the United States, compared to 3,940 (2,670 to 5,180) of the 53,100 mortalities due to aviation-attributable ozone exposure. This yields a combined estimate of 5,550 (4,290 to 6,790) premature mortalities in the US due to global, year-2015 aviation emissions. Fuel burn in the US domain therefore results in an additional 39 US mortalities per Tg of fuel burn, whereas fuel burn outside of the domain results in 20 US mortalities per Tg of fuel burn. However, aviation emissions outside the US domain cause 72% of aviation-attributable health impacts within the US domain since 83% of global fuel burn occurs outside the US.

The relative contribution of US aviation emissions to aviation-attributable surface air quality degradation in each region is shown in Figure 4 as the black segment of each bar. This is again calculated as the expected change in surface air quality if aviation emissions could be eliminated over the US. Aviation emissions from within the US domain contribute 37% and 24% of aviation-attributable exposure to PM$_{2.5}$ and ozone respectively in the US. At C24 (~400 km) resolution these contributions are higher - 49% and 30% respectively - despite total exposure being 26% and 17% lower relative to that calculated at C180 (~50 km).

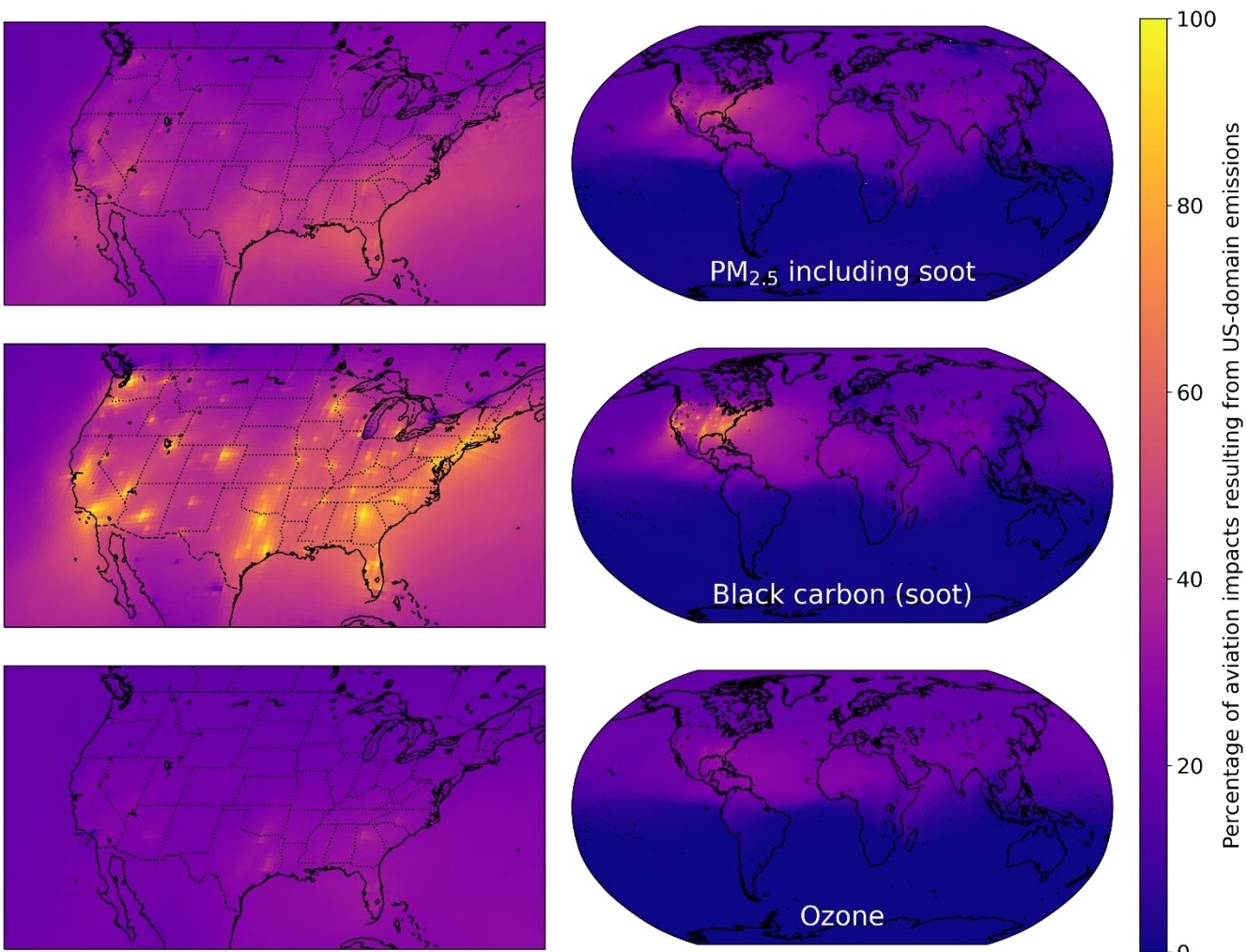

**Figure 5. Relative contribution of aviation in the US domain to all air quality impacts of aviation. Light colors indicate that emissions over the US domain are the dominant contributor, while dark colors indicate that non-US emissions are dominant. Data is shown for calculations performed at a global resolution of C180 (~50 km).**

315     The same US emissions contribute to a lower proportion (19%) of aviation-attributable exposure to $PM_{2.5}$, but a greater absolute value of 54 ng/m$^3$ in China compared to 32 ng/m$^3$ in the US. This is consistent with the hypothesis that aviation-attributable emissions are promoting the formation of $PM_{2.5}$ from non-aviation sources, rather than direct emissions of aviation $PM_{2.5}$ being responsible. Specifically, particulate matter precursor gases (from all sources) are present in greater concentrations over China than over the US, meaning that the same quantity of aviation-attributable oxidant would result in

320     more additional particulate matter. For example, the population-weighted mean mixing ratio of $NO_2$ is 2.1 times greater in China than in the US, indicating greater levels of background pollution.

The exception to this is soot. 82% of US exposure to aviation-attributable soot is the result of aviation emissions from within the US domain, whereas 6.2% of aviation-attributable exposure in China is attributable to US-domain aviation emissions.

However, aviation-attributable soot exposure in the US at C180 makes up 1.4% of total aviation-attributable $PM_{2.5}$ exposure due to the rapid wet scavenging of soot from the atmosphere (Wang et al., 2014). The reason for the greater concentration and resolution-sensitivity of aviation-attributable soot in the US is unclear.

This result is shown globally in Figure 5. As in prior figures, we show the annual average value of $PM_{2.5}$ and ozone. The relative contribution of US-domain aviation emissions to US air quality is greatest for black carbon, where it reaches nearly 100% around airports. For ozone, impacts across most of the Northern hemisphere are close to the fractional contribution of US emissions to global fuel burn, 17%. For $PM_{2.5}$ the relative contribution is more heterogeneous than for ozone but less than for black carbon. This is due to the contribution of secondary particulate matter, such as ammonium nitrate and acidic sulfate aerosols, which form in response to aviation-attributable ozone.

### 3.3    Chemistry and fate of aviation emissions by season and the role of model resolution

In order to understand the results discussed above, we also investigate the mechanism by which surface-level ozone and $PM_{2.5}$ are changed in response to aviation. Whitt et al. (2011) showed that $PM_{2.5}$ emitted or formed at cruise altitudes is unlikely to survive to the surface, and previous work has shown that the change in surface-level $PM_{2.5}$ is likely to be the result of aviation $NO_x$-attributable ozone reaching the surface and promoting formation of $PM_{2.5}$ locally from existing (non-aviation) precursor species (Eastham and Barrett, 2016). Aviation's effects on surface air quality have also consistently been found to be maximized during winter (Eastham and Barrett, 2016; Lee et al., 2013; Cameron et al., 2017; Phoenix et al., 2019). We therefore focus our analysis on seasonal changes in aviation-attributable ozone, using results from the sensitivity simulations performed with GCHP 14.2.2 (see Section 2.3).

Figure 6 shows how the vertical profile of ozone averaged from 15° to 45°N, in addition to odd oxygen production rates, is affected by aviation emissions during each season. Aviation-attributable surface-level ozone is maximized during winter and minimized during summer. This is true at all altitudes below 10 km, whereas background ozone (rightmost panel) is greatest during summer over the same range of altitudes.

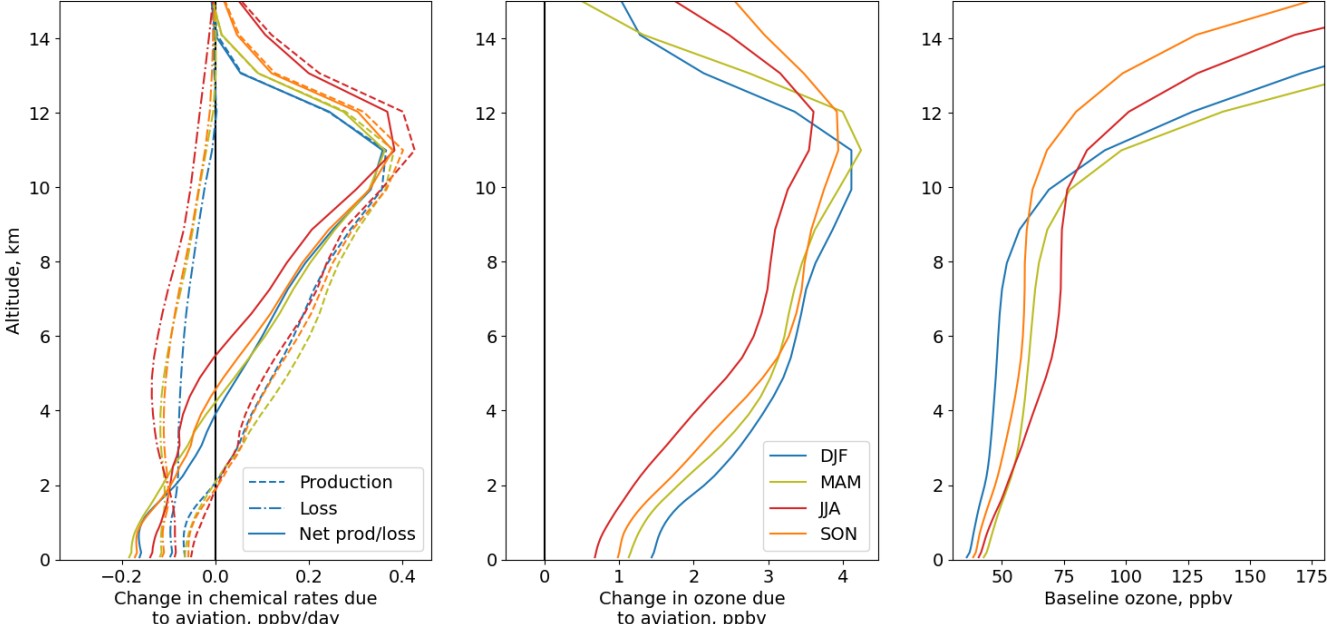

**Figure 6. Changes in production and loss rates of odd oxygen (left) and the net effect on ozone mixing ratios (centered) along with baseline ozone mixing ratios (right), separated by season. Values are for the average from 15° to 45°N at C90 resolution.**

We attribute the difference to two causes. First, the change in odd oxygen production rates in the free troposphere is consistent throughout the year, while odd oxygen loss rates are reduced during the winter. This results in longer lifetimes for aviation-attributable ozone without a decrease in production.

The second factor is transport. Figure 7 overlays the aviation-attributable change in ozone with the mean pressure (vertical) velocity in Northern Hemispheric winter (DJF) and summer (JJA), excluding sub-grid convection. As discussed by e.g.
Williams et al. (2019), the contribution of stratospheric air to surface ozone is controlled by both the degree to which upper tropospheric air is enriched by stratospheric ozone and the rate of vertical mixing from the upper troposphere to the surface. The former factor is not relevant to aviation's influence, which as discussed above shows relatively little variability between seasons in this latitude range although aviation-related increases in ozone are greater in JJA north of 45°N. The latter factor of increased transport rates, however, does appear to be significant. Downwelling motions in Northern midlatitudes during
winter months allow ozone to descend from cruise altitudes to reach the surface. When combined with reduced ozone loss rates, the result is an increase in the amount of aviation-attributable ozone reaching the surface during winter.

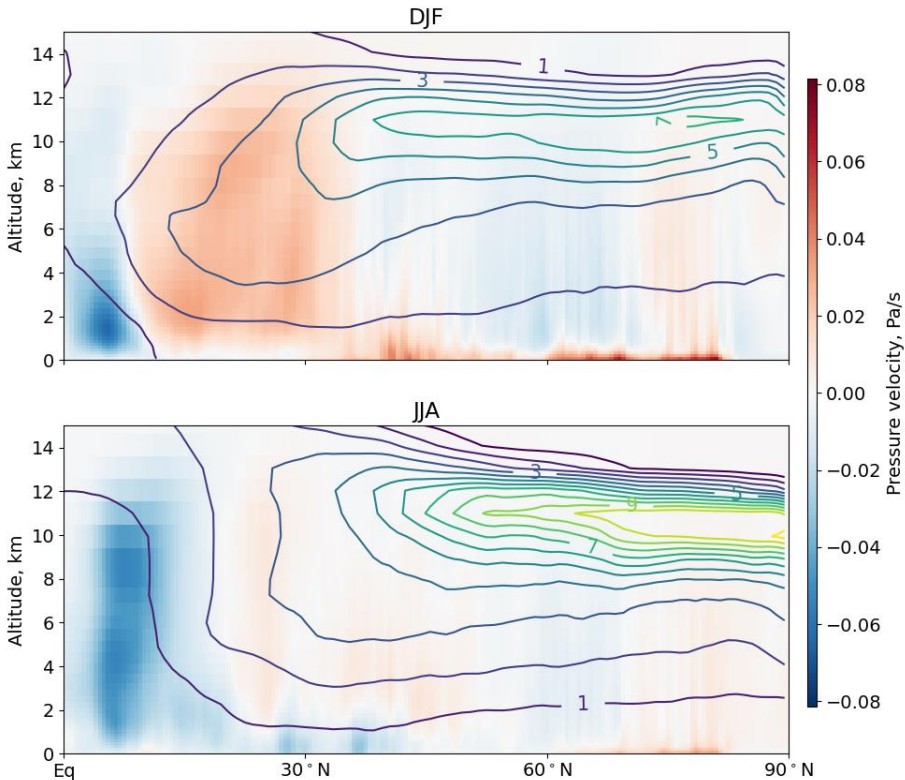

**Figure 7. Seasonal mean pressure velocities (shading, Pa/s) and aviation-attributable changes in ozone (contours, ppbv). Blue colors indicate an upwelling motion, while red colors indicate a downwelling motion. Each contour corresponds to a change of 1 ppbv ozone. Pressure velocities are extracted from the MERRA-2 meteorological data used for all simulations.**

An additional factor in the relationship between cruise altitude emissions and surface air quality is the relative timescale of vertical versus zonal transport. Based on data from the MERRA-2 reanalysis the annual mean zonal wind speed between 15 and 45°N, from 6 to 10 km altitude, is between 9.4 and 18 m/s. At these speeds an aircraft emission or its products would traverse the North American domain shown in Figure 1 at the equator within 4.9 to 9.6 days, on average. Given that aircraft cruise at a typical pressure of 300 – 200 hPa (i.e. 700 – 800 hPa from the surface) and that average pressure velocities are less than 0.1 Pa/s (or ~90 hPa/day) as shown in Figure 7, this implies that cruise-altitude emissions and their products over the US would typically leave the North American domain before they could reach the surface.

This explains why the global pattern of impacts from emissions over North America is not substantially different from the pattern of impacts from emissions over the rest of the globe (Section 3.2). During summertime, slower zonal mixing and shorter lifetimes result in US aviation emissions causing 3.9 times more ozone (population-weighted mean exposure) over the US than over the rest of the world. This ratio falls to 1.3 during winter. Since wintertime changes in ozone dominate aviation-attributable changes in surface air quality, this lower wintertime ratio dominates the overall signal such that out-of-region emissions contribute 79% of the aviation-attributable annual mean increase in US exposure to ozone.

The changes we calculate in surface air quality are predominantly due to aviation $NO_x$ emitted at cruise altitude. In a
sensitivity simulation where emissions of aviation $NO_x$ are excluded, the wintertime (DJF) aviation-attributable increase in
population-weighted mean exposure to ozone and $PM_{2.5}$ is decreased by 108 and 103% respectively. In a second sensitivity
simulation in which aviation emissions below 1 km altitude are eliminated (i.e. those associated with landing and take-off),
the decrease is 0.029 and 5.9% respectively. This is consistent with the finding by Barrett et al. (2010a), Lee et al. (2013),
and Prashanth et al. (2022) that emissions of cruise-altitude $NO_x$ are the dominant contributor to aviation-attributable
changes in atmospheric ozone and aerosol concentrations.

This leaves the question of how differences in the representation of the same physical phenomena at different resolutions
might cause discrepancies in surface-level air quality changes attributable to aviation emissions. Although the emissions data
and underlying meteorological fields are identical, simulated physical processes including transport, deposition, and
chemistry are affected by the change in resolution.

With regards to vertical transport and scavenging Yu et al. (2018) found that, when using the same model with
meteorological data from the same source as in this study, degrading the grid resolution from 0.25°×0.3125° (~C360) to
2°×2.5° (~C48) resulted in a reduction in the quantity of soluble material ($^7$Be) descending from the tropopause to the
surface. This effect was strongest in the subsiding subtropics, with a maximum effect of "up to 40%" as measured by
changes in the surface-level concentration of $^7$Be. In a sensitivity simulation performed at both C90 and C24 resolution, we
find that wintertime surface-level concentrations of $^7$Be between 15 and 45°N are 7.1% greater when simulated at C90 than
when simulated at C24. At 8 km altitude this is reversed, with concentrations 10% greater in the C24 simulation than in the
C90 simulation. This suggests that vertical mixing across the tropopause is overestimated at C24 resolution but that vertical
transport to the surface may be underestimated, and is consistent with the finding that aviation-attributable surface-level air
quality impacts are underestimated when simulated at C24 resolution. This is further supported by comparison of differences
in aviation-attributable zonal mean ozone calculated at different resolutions. Figure S3 compares the change in zonal mean
ozone due to aviation emissions calculated at C180, C90, and C24 resolution. We find that, although the change at cruise
altitudes is approximately consistent between resolutions, mixing ratios simulated at C24 are overestimated above cruise
altitudes and underestimated below them.

With regards to chemistry, we find that the changes in the model resolution cause differences in cruise-altitude
concentrations of key precursor species. At C24 resolution, baseline mixing ratios of ozone during wintertime at 10 km
altitude are 5.8% greater (from 15 to 45°N) than at C180, while concentrations of $NO_x$ and $NO_y$ are both 7.8% lower. The
opposing signs are consistent with previous studies which found an increase in short-lived ozone from aviation emissions
when simulated non-aviation $NO_x$ emissions are reduced (Holmes et al., 2011). Since we also find that surface-level ozone is
lower at C24 than at C180, this implies that differences in the representation of chemistry and background composition at
low resolution partially compensate for the reduction in vertical transport diagnosed above. At C90, these differences are
reduced to less than 1%.

### 3.4 Additional sources of uncertainty in the atmospheric response to aviation emissions

Due to the diffuse nature of aviation emissions impacts, direct empirical constraints on aviation-induced changes in air quality are more challenging than might be the case for surface-based emissions sources. As the focus of this work is on understanding the role of model resolution and the relative contribution of in-region versus out-of-region emissions we do not perform an exhaustive quantification of other potential sources of uncertainty. Nonetheless there are known sources of uncertainty which are likely to affect the simulated impacts of aviation on air quality, and we perform a brief analysis of our results in order to identify potential additional sources of uncertainty.

An evaluation of the climate impacts of aviation $NO_x$ by Holmes et al. (2011) highlighted four areas of particular concern for understanding aviation-attributable ozone changes: chemical kinetics, non-aviation anthropogenic emissions, emissions of lightning $NO_x$, and modifications to the underlying model. In addition, work by Barrett et al. (2010a) and Quadros et al. (2020) highlighted the potential role that uncertainty in aerosol emissions and aerosol modeling might have with regards to estimates of aviation-attributable changes in surface-level $PM_{2.5}$.

With these factors in mind, we compare the results calculated at a single resolution (C90) between two versions of the same model, GCHP 12.6.2 and 14.2.2 (see Section 2.3). We find that, due to changes in physical modeling of atmospheric chemistry and physics, the increase in population mortality attributable to aviation decreases by 15% between the two model versions. This is similar to a 12% change in short-lived ozone radiative forcing estimated for two different versions of the same model in Holmes et al. (2011) but smaller than the range of uncertainty reported in a multi-model intercomparison by Cameron et al. (2017). These uncertainties are in addition to uncertainties in the health response already quantified.

With regards to aerosol modeling, we find that 76% of the total increase in surface-level fine particulate matter (as calculated in GCHP 14.2.2) is nitrate by mass, compared to 22% ammonium and less than 2% sulfate or carbonaceous aerosol. The baseline composition of aerosol (excluding sea salt and dust) is 25% nitrate, 19% ammonium, and 31% sulfate. Although GEOS-Chem includes a sophisticated treatment of inorganic aerosol, uncertainty in nitrate aerosol production and concentrations remains high with several recent updates to GEOS-Chem directly affecting inorganic aerosol (Shah et al., 2023; Moch et al., 2020). This means that, alongside other physical uncertainties, the above assessment is sensitive to ongoing work to improve the representation of inorganic aerosol, and in particulate nitrate, in global models.

### 3.5 Comparison to previous work

These impacts of aviation emissions on global air quality and mortality are consistent with a recent analysis which found that year-2005 aviation emissions resulted in 58,000 mortalities globally, of which 38,000 were due to ozone exposure (Quadros et al., 2020). Our 2015 emissions inventory includes 33% more fuel burn by mass than was used for their study, and our estimate of the net mortality impact is also 33% greater, although we find that 71% of mortalities are due to ozone whereas Quadros et al. (2020) find 66%. The larger contribution of ozone in this work and Quadros et al. (2020) relative to earlier assessments is due to the use of more recent epidemiological data which includes wintertime ozone exposure (Turner et al.,

2015). This is significant because, as shown in Section 3.3, aviation-attributable ozone is maximized during winter and
minimized during summer (Eastham and Barrett, 2016). If we instead use epidemiological data from an earlier study by
Jerrett et al. (2009) which considers only summertime ozone, ozone-related health impacts of aviation emissions are
estimated to be 68% lower and net impacts 49% lower. Given that there are relatively few studies of the health impacts of
chronic exposure to ozone compared to exposure to $PM_{2.5}$, this uncertainty is a key area of future research. A more detailed
assessment of the sensitivity of our conclusions to the health impact assessment method is provided in the Supplemental
Information.

These results are also consistent with studies of aviation's air quality impacts as calculated using models other than GEOS-
Chem. A multi-model intercomparison by Cameron et al. (2017) found that global surface ozone in 2006 was increased by
0.21 to 0.65 ppbv per teragram of emitted $NO_x$ on a nitrogen mass basis (TgN). Although GEOS-Chem was included in that
study, it was neither the lower nor upper bound of the ozone response. We find a value of  0.59 ppbv per TgN, within the
range reported by Cameron et al. (2017). Similarly, Vennam et al. (2017) used a hemispheric-scale version of the CMAQ
model to investigate aviation's air quality impacts, finding an increase of 0.53 ppbv ozone per TgN. Lee et al. (2013) and
Phoenix et al. (2019) used the Community Atmosphere Model (CAM) to investigate the same question and reported an
increase of "several" and 1.78 ppbv per TgN respectively in wintertime Northern Hemispheric ozone. Similarly, the
Cameron et al. (2017) evaluation estimated increases in global surface $PM_{2.5}$ of between -210 and +96 ng/m$^3$/TgN, compared
to our estimate of +12 ng/m$^3$/TgN.

Nevertheless, our results constitute a significant increase in the estimate air quality impacts of aviation relative to studies
such as Eastham and Barrett (2016) and Barrett et al. (2010a). The former reported 6,800 premature mortalities per year due
to aviation-attributable ozone exposure and 9,200 due to aviation-attributable $PM_{2.5}$ exposure, compared to 53,100 (95% CI:
36,000 – 69,900) and 21,200 (19,400 – 22,900) respectively in this work. Assuming linear or near-linear relationships the
factor of 7.8 increase in ozone-related impacts is mostly attributable to the updated epidemiological data as described above,
which results in a 3.2 times increase in ozone-related mortality. This is accompanied by a factor 1.6 increase due to the
Turner et al. data being applicable to all respiratory diseases and not just chronic obstructive pulmonary disease and asthma,
yielding a net factor of 5.0 relative to the approach used in Eastham and Barrett (2016). An additional factor of 1.3 is due to
increased fuel burn, since our work examines 2015 whereas these previous studies examine aviation in 2006. We also
estimate that the use of finer resolution of our analysis (~50 km) compared to the ~500 km resolution of the previous studies
results in an increase in the estimated ozone-related impacts of aviation emissions by a factor of 1.2. These three factors
combined imply a 7.6 times increase in ozone-related impacts, consistent with the factor of 7.8 which is observed.

For $PM_{2.5}$ we estimate 2.3 times as many mortalities as were estimated by Eastham and Barrett (2016). This smaller factor is
only partially explained by the factors described above. While the same factor of 1.3 applies for fuel burn, the concentration
response function used here results in 14% fewer mortalities being attributed to aviation emissions than if the approach used
in the prior study is applied (Hoek et al., 2013). The greater resolution of our work increases the estimated $PM_{2.5}$-related
mortality by a factor of 1.35, but combined these result in a factor of 1.5 compared to the observed factor 2.3 increase.

The remaining factor 1.6 difference in $PM_{2.5}$ mortality between the two studies may be in part due to growth in non-aviation emissions between 2006 and 2015. For this work we use emissions from 2014 in the Community Emissions Data System (CEDS) (Hoesly et al., 2018) as a proxy for anthropogenic, non-aviation emissions. Figure 2 shows that the highest aviation-attributable concentrations of $PM_{2.5}$ are found in Asia, and analysis of CEDS data suggests that emissions of NO and $SO_2$ from non-aviation sources in Asia (defined as a region bounded by 60 to 150°E and 10°S to 55°N) increased by 20% and 0.16% respectively between 2006 and 2014. In the same period, NO and $SO_2$ emissions fell by 5.3% and 16% respectively outside of Asia. This change was accompanied by a 9.4% increase in ammonia emissions in Asia, and a 7.3% increase outside. Such changes would increase ambient concentrations of $PM_{2.5}$ precursors, and therefore increase the amount of $PM_{2.5}$ formed as a result of aviation-attributable ozone descending to the surface. Other possible contributors to the increase in $PM_{2.5}$ exposure-related mortality attributable to aviation include changes in baseline mortality rates and increases in the exposed population. For example, using the same geographical boundaries as were applied to calculate changes in Asia's emissions from CEDS, the total population in Asia increased by 11% from 2005 to 2015. A simple linear combination of increases – 20% in NO, 9.4% in ammonia, and 11% in population – would imply a factor 1.45 increase in exposure. However a detailed analysis would be required to fully understand the sources of the unexplained factor 1.6 difference. An additional potential cause of difference is the meteorological year or data used for each study. Barrett et al. (2010a), Eastham and Barrett (2016), and Quadros et al. (2020) each investigated the degree to which a change in meteorology could affect the impacts of aviation on air quality, finding differences of between 5 and 21% in estimated population exposure and health impacts.

## 4    Discussion

This work finds that aviation's air quality impacts are greater than has been previously estimated. Since prior assessments have shown aviation's monetized impacts on air quality to be similar in magnitude to its impacts on the climate (Grobler et al., 2019), this work suggests that impact mitigation options which do not address $NO_x$ emissions-attributable air quality impacts will therefore not address one of the largest environmental impacts of aviation. This may change the balance of cost-effectiveness when considering trade-offs between $CO_2$ emissions and $NO_x$ emissions.

These results also imply that coarse-resolution global simulations may underestimate the impacts of aviation on surface air quality. However, moderate-resolution studies at C90 (around 100 km or 1° resolution) resolve 98% of the impacts calculated by a C180 simulation at one quarter of the computational cost (or one eighth if a smaller time step is required). This includes 99% of $PM_{2.5}$ impacts and 97% of ozone impacts. This work does not address the degree to which different parameterizations of physical processes might affect estimates of aviation impacts, as might be relevant when comparing results from regional or local-scale air quality models to those from global-scale models.

This work explains differences in the literature regarding aviation's air quality impacts. Yim et al. (2015) found that the use of finer resolution nested modeling increased estimated ozone exposure by 12% but decreased exposure to $PM_{2.5}$ by 29%,

whereas Vennam et al. (2017) found reductions of more than 90% in both. We hypothesize that the differences found between simulations at different resolutions in the former study are due not to unresolved local-scale processes (Lee et al., 2023) but rather to their use of two inconsistent models to represent the global and nested regions, as the difference in results is within the range reported in a model intercomparison of aviation's air quality effects (Cameron et al., 2017). Vennam et al. (2017) instead used boundary conditions from a single global simulation which included aviation emissions for both their aviation and non-aviation regional simulations of North America, inherently removing the effect of aviation $NO_x$ on hemispheric-scale tropospheric ozone and therefore neglecting the influence of a larger atmospheric response. Our work shows that this global response drives the majority of the change in surface air quality. Not only do we find that each kilogram of fuel burned outside of US air space still cause about half as many mortalities per kilogram as fuel burned over the US, but we show that the vertical mixing timescales needed for cruise altitude emissions to affect surface concentrations are greater than the zonal mixing timescales (Section 3.3). This implies that a model with fixed boundary conditions over a single region cannot capture the influence of cruise altitude emissions – even within the same region – on surface air quality.

This work describes the potential benefit of a rapid reduction in aviation emissions, and the degree to which different modeling approaches can accurately capture the expected outcome of such a reduction while accounting for non-linearity in the response. Approaches which are intended to perform an attribution of current-day air quality or mortality impacts between different sectors, regions, or species using methods such as tagging (Emmons et al., 2012; Butler et al., 2018) may find different results. As discussed by Clappier et al. (2017) and Thunis et al. (2021a, b), the relevance of these results to planned policy will therefore depend on the context and objectives of the policy.

A remaining important uncertainty regards the response of human health to ozone exposure. Although several recent epidemiological analyses have findings consistent with Jerrett et al. (2009) and Turner et al. (2015) that ozone exposure increases morbidity and mortality (Zhao et al., 2021; Lim et al., 2019; Rhee et al., 2019), including at low concentrations (Yazdi et al., 2021), other studies have suggested that ozone exposure may not be significantly associated with mortality (Atkinson et al., 2016; Brunekreef et al., 2021; Huangfu and Atkinson, 2020). This would not affect the finding that over 20,000 mortalities per year due to $PM_{2.5}$ exposure are attributable to aviation.

There has also been research suggesting that exposure to certain specific components of $PM_{2.5}$ may be more harmful than other constituents of $PM_{2.5}$, which may indicate greater impacts related to soot, ultrafine non-volatiles, or organic carbon (Verma et al., 2015). If so, our finding of a greater relative sensitivity of soot exposure to model resolution suggests that the more localized impacts of aviation soot emissions will require higher-resolution simulations or localized modeling approaches such as those in Yim et al. (2015) to quantify.

## 5    Conclusions

Our findings show that 74,300 (95% confidence interval due to uncertainty in health response: 57,300-91,100) premature mortalities each year are attributable to aviation emissions, based on 2016 data. This is 4.6 times greater than a previous

assessment for 2005 finding 16,000 premature mortalities each year. A factor of 1.29 is due to our use of a global model with greater spatial resolution; a factor 1.84 is due to the availability of an ozone concentration response function which includes wintertime ozone; a factor 1.24 is due to the inclusion of a broader set of diseases in those impacted by ozone exposure; and a factor 1.28 is due to increases in aviation fuel burn. Accounting for the different effects each of these factors has on ozone and $PM_{2.5}$-attributable mortality, these alone explain a factor 4.1 increase in the estimate, 87% of the total. We hypothesize that the remaining 13% discrepancy may be the result of regional increases in non-aviation emissions.

We find no evidence of a decrease in air quality impact with increasing model resolution. Instead we find that the simulated impacts of aviation emissions on surface air quality increase by 24% when using a 50 km model compared to a 400 km resolution model. Impacts simulated at a resolution of 100 km globally are within 2% of those at 50 km, suggesting that moderate-resolution simulations are capable of accurately simulating aviation's air quality impacts.

Finally, we show that the impacts of aviation on air quality are global in nature. Emissions from aircraft flying over the US cause 39 mortalities per Tg of fuel burn in the US, compared to 20 mortalities in the US per Tg of fuel burned outside this region. We also find that $PM_{2.5}$ concentrations in China are more strongly affected in absolute terms by aviation emissions within the US domain than concentrations of $PM_{2.5}$ in the US, and that concentrations of aviation-attributable $PM_{2.5}$ are more strongly correlated with the concentration of non-aviation $PM_{2.5}$ than with a marker of local aviation emissions.

Previous work has shown that the monetized air quality impacts of a unit of aviation fuel burn are similar in magnitude to aviation's monetized climate impacts, including effects of contrails. The further increase in estimated air quality impacts we find suggest that aviation-attributable air quality degradation is a significant contributor to aviation's environmental impacts, and that mitigation of full-flight $NO_x$ emissions should be considered alongside ongoing efforts to reduce the effects of aviation on the climate.

**Acknowledgements and funding**

The MERRA-2 data used in this study have been provided by the Global Modeling and Assimilation Office (GMAO) at NASA Goddard Space Flight Center. The core computations in this paper were run on the FASRC Odyssey cluster supported by the FAS Division of Science Research Computing Group at Harvard University. Sensitivity simulations were performed using the Svante cluster at the Massachusetts Institute of Technology. This work was partially supported by the U.S. Federal Aviation Administration (FAA) Office of Environment and Energy through ASCENT, the FAA Center of Excellence for Alternative Jet Fuels and the Environment, project 58 through FAA Award Number 13-C-AJFE-MIT under the supervision of S. Daniel Jacob and Jeetendra Upadhyay. Any opinions, findings, conclusions or recommendations expressed in this material are those of the authors and do not necessarily reflect the views of the FAA.

**Code availability statement**

The core of this work was completed using version 12.6.2 of the GEOS-Chem High Performance (GCHP) code. The specific codebase used can be found at https://github.com/geoschem/gchp_legacy/releases/tag/12.6.2. Additional sensitivity simulations were performed using version 14.2.2 of the same code as described in Section 2.3.

575 **Author contributions**

SDE, RLS, and SB conceived and designed the study. GC generated the aircraft emissions data. SDE performed all model simulations and analysis. DJJ provided computational resources for high-fidelity simulation. All authors contributed to the manuscript writing and editing.

**Competing interests**

580 The authors declare no competing interests.

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
