# Peer review of "Global impacts of aviation on air quality evaluated at high resolution"

_EGUsphere, 2023_

## Author Comment (AC1)

**Dr. Sebastian D. Eastham**
Principal Research Scientist
Joint Program on the Science and Policy of Global Change
Associate Director of the Lab. for Aviation and the Env.

[Figure]

**Massachusetts Institute of Technology**
77 Mass. Ave. Office E19-439F, Cambridge MA 02139, USA
http://globalchange.mit.edu | http://lae.mit.edu
seastham@mit.edu | (617) 253-2170

Editorial Office

Atmospheric Chemistry and Physics

November 14th, 2023

Dear Dr. Querol and reviewers,

**Revised submission of "Global impacts of aviation on air quality evaluated at high resolution" to _Atmospheric Chemistry and Physics_**

Thank you for your time and effort in considering our manuscript. We greatly appreciate the comments made by the reviewers, and have made significant efforts to address their concerns.

We recognize that a broad concern expressed by the reviewers regarded the robustness and interpretation of our findings. In response we have now performed an additional set of simulations, re-evaluating the response of surface air quality to aviation emissions but now including additional sensitivity scenarios and diagnostic data which have allowed us to dig more deeply into the root causes of the differences we first observed. These include:

- A full-length simulation using an updated version of the base model, to evaluate sensitivity to bulk changes in our understanding of atmospheric science and emissions;
- An additional series of tracer transport tests to provide quantitative evidence for claims regarding vertical mixing times between seasons; and
- Two simulations extending through northern hemispheric winter (the time of peak exposure to air quality changes resulting from aviation emissions) seeking to evaluate:
  - The degree to which $NO_x$ emissions alone can be said to cause these impacts (removing the possibility of near-airport primary $PM_{2.5}$ as being the driving cause); and
  - The relative contribution of landing and take-off emissions (below one kilometer altitude).

Due to the high computational cost of these simulations we have conducted our sensitivity analyses at C90 (~100 km) resolution, which our earlier simulations showed are sufficient to resolve ~98% of air quality impacts associated with aviation in a higher-resolution (C180, ~50 km) simulation. In each of these simulations we have archived and analyzed additional data in order to better understand and communicate the impacts of aviation on air quality.

In addition, and guided by the reviewers' comments regarding the need for more scientific insight into aviation emissions impacts, we took this opportunity to archive a greater variety of information from the additional simulations. This has allowed us to evaluate: the long term fate of aircraft emissions compared to their products; where, and the degree to which, ozone production and loss rates are changed by aviation emissions; and the potential contribution of differences in background atmospheric composition at different resolutions.

Finally, we realized during this review process that an additional post-processing step was applied internally when simulating the effect of eliminating aviation emissions over the North American region, but that this step had not been described in the manuscript. This step means that emissions were eliminated over US land specifically (at all altitudes) rather than in a simple block domain. This does not substantially affect the meaning of the manuscript, which is focused on the effect of resolution and the relative contribution of near-airport emissions. However, it does increase the relative contribution of in-domain emissions. Nonetheless the central point – that 72% of aviation-attributable impacts on US air quality come from out-of-domain emissions – is unaffected. We have also changed the manuscript discussion throughout to reflect this, more accurately referring to the "NA" domain as instead the "US" domain.

We believe that the changes we have made to the manuscript have resulted in a substantial improvement. In addition to these bulk changes, we have also made specific additional changes to address each of the reviewer

comments. A response to each reviewer comment is given below; the original comments are shown in **bold** with our responses shown in *italics*.

***Referee #4***

**The authos use the GEOSChem model to run 1-year simulation with and without aviation emissions to attibure mortalities to the changes in ozone and PM2.5. They use different resolutions and calculate the differences aviation related mortalities. In general the paper is well written, however, a deeper analysis that tries to explain differences in the modelling results due to resolution is lacking. Moreover, I have severe doubts concerning the method of an incremental approach that is applied in this case. Is the sum of all incremental approaches for all individual sources, such as industries, households, transport, aviation, etc. giving the total mortalities due to either ozone and PM2.5? Thunis et al. (2019) and the refernces therein clearly show that this is not the case. They have a very strong argument that incremental approaches can not be used for source (and here mortality) attribution. I think addressing these points is crucial before the paper may become accepted.**

*We thank the reviewer for their comments and their thorough review. We have taken these concerns seriously and have committed significant resources to ensure that we are able to provide quantitative data which more completely analyze the underlying causes of our findings. We have also worked to ensure that the goal of our approach is not conflated with a broader assessment of source attribution, but rather focused specifically on the question of the avoidable impacts of aviation, what fraction of these impacts result from in- or out-of-region emissions, and the degree to which assessments of these quantities might be sensitive to model resolution. A detailed response to each comment is provided below.*

**I have some doubts whether the approach in calculating aviation related mortalities is applicable. Around line 160: "This is the quantity for which Turner et al (2015) determined a 12% increase (95% confidence interval: 6.0-18%) in respiratory mortality per 10 ppb increase in ozone exposure."  To my understanding this analysis is based on situations where an enhancement of ozone can be attributed to an increase in an emission source. However, here, we have a constant emission of NOx by aviation that competes with other sources of NOx, such as industry, lightning etc. In that case Thunis et al. 2019 (and the literature cited therein) clearly pointed out that an increamental approach, with and without aviation emissions in not applicable. If the mortalities due to ozone are estimated with your approach for all individual emissions separately and added up would that give the same number as for all emissions? It is a highly non-linear system, so they answer is not (see also Tunis et a.).   I think that the uncertainties due to the used method are much larger than due to different resolutions**

*We fully agree that an approach aimed at understanding the fraction of current air quality degradation which is attributable to different sources, sectors, or regions might find a different answer. However, our objective here is to quantify a) what the potential improvement of air quality might be if aviation emissions could be reduced to zero, b) what fraction of this benefit can be achieved in one region (the US) through actions taken only in that region, and c) the degree to which these quantities are sensitive to model resolution (and therefore the degree to which policy assessments relying on a coarse-resolution model might over- or under-estimate potential benefits). As discussed elegantly by Clappier et al. (2017), the choice of which approach to use is sensitive to the question being asked. For example, it is entirely possible to find that two different sectors could, if eliminated independently, reduce air quality mortality by 35%, but that performing each together could achieve a total of only 50% (as discussed in e.g. Thunis et al. (2021)). This does not change the fact that eliminating a single sector would still reduce total air quality impacts by 35%, but it does make it important for readers (in particular policy makers) to understand the role of non-linear interactions. We now explicitly call out this need for nuance and sensitivity to context in the discussion (lines 512-517).*

**The analysis of the impact of different resolutions lacks quite some analysis.**
**a) How are regional pattern e.g. around major airports changed? and why?**

*We now include a discussion of this effect as part of Section 3.1 (lines 283-293) which shows how concentrations of black carbon are increased local to US airports in such a way that simulation at C180 is more accurate, but that the changes in ozone and PM$_{2.5}$ are less strongly localized (see $\mathrm{Figure\ R\ 1}$ below, which is now also included in the SI as Figure S2). As illustrated by the relatively small differences in mortality between simulations at C90 and C180, simulations at these resolutions demonstrate qualitatively similar patterns of changes in PM$_{2.5}$ and ozone but neither are dominated by proximity to large airports. The same is not true for black carbon (soot), which shows strong peaks near to major airports (e.g. the hot spot near LAX and Santa Monica Airport) which are poorly resolved*

*at coarser resolutions. However, even in this hotspot soot makes up less than 3% of total PM$_{2.5}$ exposure, meaning that the overall health impact calculated for aviation emissions is not significantly affected by the transition from C180 to C90. We do however now highlight that soot specifically is more accurately captured at C180, such that studies focused on near-airport air quality (as opposed to regional or global air quality) would benefit for very high resolution simulations which are not plausible for global analyses.*

[Figure]

*Figure R 1. Changes in annual mean PM$_{2.5}$ (left), soot (center), and MDA8 ozone (right) over the US west coast due to aviation simulated at C180 (top),C90 (middle), and C24 (bottom) resolution. Red dots indicate airports which served at least 100,000 passengers in 2015 according to the US Bureau of Transportation Statistics from T-100 segment data.*

**b) How is the transport changed due to resolution changes? Convective transport, lighning? Scavenging?**

*Bulk (advective) transport is in theory unchanged, as we are using a chemistry-transport model where meteorology is read in as a set of externally-imposed fields. However we still expect some effect due to (e.g.) the inability of a coarse model to resolve local eddies, including convective motions which may have only been resolved at fine spatial and/or temporal resolution. These phenomenon are discussed specifically in the context of this model in Yu et al. (2018).*

*For scavenging it is possible that there is some resolution dependence, although the cloud data (including cloud fraction, water content, and precipitation rates) are again pre-calculated in the meteorological data. As such we would not expect significant non-linearity. Unfortunately we do not have scavenging data available from our original simulations, which would otherwise have allowed us to evaluate the degree to which (e.g.) aviation-attributable NO$_y$ is washed out differently at different resolutions.*

*Finally, lightning emissions are calculated internally based the cloud convective depth and flash density. These fields are archived at a single, common resolution in order to minimize differences in lightning emissions between resolutions. As such, no significant differences are expected between simulations at different resolutions. This is now stated in the Supplementary Information.*

*We now discuss these components of resolution dependence in in Section 3.3 (lines 384-401 specifically), which is dedicated to providing insights into both the base mechanisms by which aviation affects surface air quality and the reasons that these mechanisms might be different at different resolutions.*

**c) Where is actually ozone production and loss rates changed?**

*We now evaluate changes in ozone production and loss rates as part of Section 3.3. The relevant text is reproduced below:*

*As shown in* Figure R 2*, the vertical profile of ozone averaged from 15° to 45°N, in addition to odd oxygen production rates, is affected by aviation emissions during each season. Aviation-attributable surface-level ozone is maximized during winter and minimized during summer. This is true at all altitudes below 10 km, whereas background ozone (rightmost panel) is greatest during summer over the same range of altitudes.*

[Figure]

*Figure R 2. Changes in production and loss rates of odd oxygen (left) and the net effect on ozone mixing ratios (centered) along with baseline ozone mixing ratios (right), separated by season. Values are for the average from 15°N to 45°N at C90 resolution. This is now Figure 6 in the main text.*

*We attribute the difference to two causes. First, the change in odd oxygen production rates in the free troposphere is consistent throughout the year, while odd oxygen loss rates are reduced during the winter. This results in longer lifetimes for aviation-attributable ozone without a decrease in production.*

*The second factor is transport.* Figure R 3 *overlays the aviation-attributable change in ozone with the mean pressure (vertical) velocity in Northern Hemispheric winter (DJF) and summer (JJA). As discussed by e.g. Williams et al. (2019), the contribution of stratospheric air to surface ozone is controlled by both the degree to which upper tropospheric air is enriched by stratospheric ozone and the rate of vertical mixing from the upper troposphere to the surface. The former factor is not relevant to aviation's influence, which as discussed above shows relatively little variability between seasons in this latitude range although aviation-related increases in ozone are greater north of 45°N. The latter factor of increased transport rates, however, does appear to be more significant. Downwelling motions in Northern midlatitudes during winter months allow ozone to descend from cruise altitudes to reach the surface. When combined with reduced ozone loss rates, the result is an increase in the amount of aviation-attributable ozone reaching the surface during winter.*

[Figure]

*Figure R 3. Seasonal mean pressure velocities (shading, Pa/s) and aviation-attributable changes in ozone (contours, ppbv). Blue colors indicate an upwelling motion, while red colors indicate a downwelling motion. Each ozone contour corresponds to a change of 1 ppbv. This is now Figure 7 in the main text.*

**I think the intercomparison with other studies mostly relies on the same model, right? What about other aviation simulations for NOx and aerosols?**

*Several prior studies of the impacts of aviation on global premature mortality have indeed used GEOS-Chem, including (for example) Barrett et al. (2010), Eastham and Barrett (2016), and Quadros et al. (2020). Yim et al. (2015) found comparable results when using the CMAQ model in local regions, but arguably their work still relied on GEOS-Chem for boundary conditions. However, there are other studies which found similar results without using GEOS-Chem. The global simulation performed in Vennam et al. (2017) found changes in surface-level pollutant concentrations which were within the range of the aforementioned studies. Similarly, the multi-model intercomparison by Cameron et al. (2017) found agreement among multiple chemistry transport models (including GEOS-Chem) regarding the magnitude of the change in surface-level ozone and $PM_{2.5}$ resulting from aviation emissions. Studies by Lee et al. (2013) and Phoenix et al. (2019) used the Community Atmosphere Model (CAM) versions 3 and 5 respectively to assess the impacts of aviation on surface air quality, finding again that aviation emissions cause increases in surface ozone of the order of 1 ppbv during winter with the dominant contribution from cruise-altitude $NO_x$. These studies are now cited in the manuscript on lines 441-450.*

**Eq. 1 and 2 (in addition to the above comments): I am wonndering, although knowing that this approach has been used frequently in the past, if this eq. is actually well representing the aviation induced mortality. We have many sources for PM2.5 and ozone such as households, industry etc. My understanding is that the mortality from all sources is M_Base. Hence the sum of all mortalities dM_i from individual sources i (aviation, households, fires, ...) should give M_Base.**

**M_Base=Sum_i dM_i=Sum_i M_Base * (RR_i-RR_Base)/RR_Base = M_Base/RR_Base* Sum_i (RR_i-RR_Base)**

**However, in general, the sum of all changes of the surface concentration and exposure is not base concentaiton or exposure, respectively. (see e.g. Thunis et al. 2019 and the literature therein). Thunis et al cearly indicated the inaccuracy of incremental approaches. Emmons et al. (2012) showed a differences of a factor of 2-4 for a surface source in using contribution and incremental apporaches. Local conditions the differences might be even larger. And the nonlinear function in eq. (2) is adding to the discrepancy. Hence, to my understanding, here the mortality changes by switching off aviation is investigated and not the impacts from aviation. Please make this clear in the text and title.**

*We agree with the assessment that we here perform a calculation of the number of mortalities which could be avoided if aviation emissions were to be brought to zero. We also agree that, for studies which seek to attribute all air quality impacts to different sectors, other approaches (such as tagging) may be more appropriate. However our intent here is to quantify the degree to which avoidable mortalities result from current-day aviation emissions, and therefore the benefit which might be achieved if those emissions could be eliminated (either through policy means or technological means, such as the post-combustion emissions control discussed in Prashanth et al. (2020)). To make this as clear as possible, we now discuss this nuance both in the introduction (lines 81-83) and the discussion (lines 512-517), stating unequivocally that the calculations performed here relate to the context of the benefit which is theoretically achievable by acting on aviation emissions in isolation. In the discussion in particular we now give the following statement:*

*"This work describes the potential benefit of a rapid reduction in aviation emissions, and the degree to which different modeling approaches can accurately capture the expected outcome of such a reduction while accounting for non-linearity in the response. Approaches which are intended to perform an attribution of current-day air quality or mortality impacts between different sectors, regions, or species using methods such as tagging (Emmons et al., 2012; Butler et al., 2018) may find different results. As discussed by Clappier et al. (2017) and Thunis et al. (2021a, b), the relevance of these results to planned policy will therefore depend on the context and objectives of the policy."*

**Eq. 1/2 Are you evaluating the exposure and mortality on the basis of, e.e., daily values and then averaging or yre you using the mean values in the first place? How large are the differences between the two approaches? So day-by-day variations in the transport pattern and e.g. chemical responss might be large enough to considerably change the exposure as a mean.**

*We calculate mortality based on the annual-average value of $PM_{2.5}$ in μg/m³, and the annual-average value of the maximum daily 8-hour average ozone concentration in ppbv. This is necessary as the epidemiological data supporting the relationship between exposure and mortality correlates an increase in long-term average (chronic) exposure with mortality risk. We do not consider the effects of short-term exposure, for which a different set of concentration response functions and epidemiological data would be required. We now explicitly state this distinction in the methods (line 174-176).*

**213 "This provides additional evidence" I wouldn't call a missing correlation as an evidence. I suggest to rewrite by using a statement such as "not contradicting to ..."**

*We agree with this assessment. The wording has been changed to "This is consistent with the hypothesis that…" (line 255).*

**228 "The greater resolution has two effects: physical phenomena are more finely resolved .." While I agree in general with the statement, I think there is more analysis required to understand the impact of resolution on physical processes. Are you sure that the simlaiton is more realistic? HOw is the vertical transport changed? Do yo have an anaylsis of convective up- and downward transports that are changed. Is the dynamical lifetime affected, and if, by which process. Are there measurements to allow a judgement on the quality of the higher resolved proesses? E.g. 222Rn? Are natural processes and sources changed? e.g. llghtning NOx? I think a much deeper analysis os required to allow a more detailed judgement.**

*We now include a discussion (Section 3.3) which focuses on understanding first the reason for the changes we observe (see discussion in response to the comment regarding ozone production rates), but also the likely effect of a change in resolution on process representation. This includes not only a concise review of relevant literature with regards to the representation of convection and transport at different resolutions with this specific model (GCHP), but also the results of additional tracer transport simulations which we have conducted.*

*With regards to changes in natural sources, lightning emissions are calculated from flash rates and convective depths which were archived at the native resolution of the meteorology specifically in order to avoid changes in resolution affecting emissions. We also use pre-calculated emissions for natural sources of dust, biogenic VOCs, and sea salt. Emissions of iodine from sea salt are still calculated online but are assumed to have a negligible effect on aviation-attributable changes in air quality. These are detailed in the Supplemental Information although we would be happy to move this description to the main text if preferred.*

**246 "The relative contribution of NA aviation emissions" plese rephrase and see above the comment based on Thunis et al.**

*We have now clarified that these calculates reflect the expected change if aviation emissions are brought to zero (lines 303-304).*

**Sect. 3 in general: The simulation is based on one year. The year 2015 is known for intense heat waves. Is there any information available on the robustness of the results, e.g. annual variability, etc.**

*We now cite in the main text (lines 481-485) estimates of meteorological sensitivity from three previous studies of the effect of aviation emissions on surface air quality, each of which evaluated the effect of the meteorological year. Barett et al. (2010), Eastham and Barrett (2016), and Quadros et al. (2020) each evaluated the effect that a change in meteorological year would have. Barrett et al. (2010) found that the choice of year and meteorological product could change estimated mortalities by up to 21%. Eastham and Barrett (2016) found that, across five different years, global exposure to $PM_{2.5}$ and ozone could vary by up to 5% due to the specific choice of meteorological year. Meanwhile Quadros et al. (2020) simulated how a change in the simulation year (reflecting changes in both background emissions and meteorology) from 2005 to 2013 would affect exposure, finding a net increase of 6.6 to 12% depending on the region of interest. We cite the range 5 to 21% as a conservative estimate.*

**Intercomarison: As far as I see most citations refer to the use of GEOSchem. right? What about other models? In general amore detailed explanation is missing.**

*We focus our comparison on previous GEOS-Chem-based studies as this allows us to more exactly separate the roles of different changes. However, we do now include a comparison of our estimated changes in surface concentrations against the multi-model intercomparison performed by Cameron et al. (2017) and against the global results from Vennam et al. (2017). We do not include a comparison against the nested results from Vennam et al. (2017) as this work suggests that the exclusion of changes in large-scale atmospheric composition in that work was the primary cause of their simulated reduction in ozone and $PM_{2.5}$ concentrations when performing a fine-resolution calculation. However, we do now refer to simulations of aviation's air quality impacts performed using other models (notably CAM 3, CAM 5, CMAQ, GEOS, GATOR-GCMOM, and GISS Model-E2) (lines 441-450).*

**After reviewing this manuscript, I do not think in its present form it is appropriate for ACP. The manuscript runs a chemical transport model at different resolutions to determine the change in near surface ozone and 2.5 micron Particulate Matter (PM2.5), and then presents 'health effects' as premature mortality. I think too much of the uncertainty and analysis is due to epidemiological uncertainty which really cannot be assessed in ACP. If the paper were to focus on the physical modeling it would be fine: but there is zero uncertainty estimated there. I think the manuscript needs to be a much better analysis of the physical modeling for ACP. If the focus is to be on the health effects and uncertainties, it should go in a health focused journal. I do not like the use of headline numbers of thousands of deaths, when there is no uncertainty in the abstract, values have changed by a factor of 5 from previous estimates, and some of the work is based on a single health study. As a physical scientist, I am not able to assess whether these are valid or not, and hence I do not think ACP is appropriate. So ideally, this would be put in a health oriented journal. If the authors wish it to be in ACP, I think it should focus on the changes to Ozone and PM2.5. Specifically there needs to be a better assessment of uncertainty, which probably means using more than one meteorological year.**

*We appreciate this perspective and have worked to address it. Firstly, we believe that there is unfortunately a need to find a common metric for impacts from different pollutants which necessitates the translation of exposure, which is species-specific, into health risks, which are not. The use of premature mortalities has also historically been supported by ACP, as evidenced by the ongoing and frequent publication of studies in ACP which quantify impacts from $PM_{2.5}$, ozone, or both in their abstract, including several studies since 2021 alone (Zhan et al. 2023; Nault et al. 2021; Geels et al. 2021; Sun et al. 2021; Zhang et al. 2021; P. Wang et al. 2023; Tarín-Carrasco et al. 2022). The specific concentration response functions which we have used have already been applied in the context of aviation (Quadros, Snellen, and Dedoussi 2020) and we do include evaluations of our results – and implications – if one assumes different concentration response functions. However we recognize that this needs to be more clearly presented in the paper, and now highlight in the abstract that this result is sensitive to the choice of CRF (line 28). We also now clearly state the 95% confidence interval in impacts resulting from uncertainty in the health response in both the abstract and discussion and close our abstract with a statement regarding uncertainty in the physical response to aviation emissions (line 29).*

*We have also made a concerted effort to improve and extend our assessment of physical uncertainty. We have performed an additional set of simulations in which we use the same set of aviation emissions with a different model version, including substantial new updates to the treatment of tropospheric chemistry (see Section 2.3). Using information from this new simulation set and from literature data we now explore the degree to which different models, different versions of the same model, different meteorological years, and different model resolutions can affect estimated exposure to both ozone and $PM_{2.5}$ and therefore health risks, with a new section (3.4) dedicated to the question of physical uncertainty.*

*Finally, we have substantially expanded our overall analysis. Previously we included two sections in the results: a combined analysis of the effect of resolution and in-region vs out-of-region emissions (3.1), and a comparison to previous work (3.2). We have now expanded this into five sections. Section 3.1 is dedicated only to quantifying the effect of resolution change and includes an evaluation of the role of near-airport emissions. This has allowed us to separate out the role of in-region vs out-of-region emissions in Section 3.2. Section 3.3 introduces a new analysis of the chemistry and fate of aviation emissions by season, based on the new C90 (~100 km) sensitivity simulations which were performed in response to this review. It also includes an analysis of the potential causes of differences between simulated impacts, based on both the results of our model simulations at different resolutions and information available from the literature, and discusses the potential causes of our findings regarding in-region vs out-of-region emissions (lines 360-372). Section 3.4 investigates two additional sources of physical uncertainty, while also placing our observed effect of a change in model version alongside literature evidence for model-based uncertainty. Finally, we have expanded Section 3.5 (formally 3.2), the comparison to previous work, to include a larger set of studies using models other than GEOS-Chem to evaluate aviation's effects on surface air quality. These are described in more detail through our point-specific responses below, but we hope that the reviewer agrees that we have significantly deepened the relevance and depth of our manuscript thanks to their suggestions.*

**I also do not think the 'different resolutions' is particularly interesting since the meteorology is just interpolated to drive the model, and because the scale of resolutions is still within the range of using exactly the same parameterization methods. This might also not be a wise focus for the manuscript as it is not particularly strong.**

*We respectfully disagree with this assessment but have worked to make the argument more compelling in the manuscript itself. One of the motivations of our work was the realization that in this model (as in other chemistry transport models), the change in resolution can significantly affect the result of the calculation specifically decspire there being no change in underlying physical parameterizations. For example, the treatment of convection within the model is based on that of the parent GCM – however the parent GCM is resolving a much larger fraction of convection in its large scale transport fields. Since this is not recovered when the meteorology's resolution is degraded, convection can be under-represented in coarse CTMs (see eg. Yu et al. (2018), now discussed on lines 384-393). However atmospheric scientists have limited resources and cannot afford to perform all their simulations at the parent model resolution; for example, the GEOS-Chem High Performance CTM used in this research requires around 35,000 CPU hours per simulation year if running at C90, but around eight times that per simulation year at C180. Finding that, in contradiction to previous work, a C90 simulation is adequate to resolve ~98% of impacts resolved at C180 (but that a much cheaper C24 simulation is not) enables researchers to more efficiently explore the causes of – and potential solutions to – aviation's environmental impacts. In light of the comments here we have endeavored to make this clear by deepening our investigation of the differences in physical modeling between simulations at different resolutions (Section 3.3).*

**Specific comments:**

**Page 1, L27: What is the uncertainty on these numbers? It seems that the uncertainty is entirely due to epidemiological factors? How much is due to ozone changes? Any of it? I am not convinced that this is useful at all given that air quality is highly non-linear and the extremes are not well reproduced by any scale of models 50-400km. Are the ozone numbers different? Suggest that things be focused around ozone and NOx quantification rather than the epidemiology.**

*We agree that extremes are not well represented by global-scale models. However, the change in surface ozone concentration due to aviation emissions is diffuse, as shown in Figure 2 of the main text, and analyses such as Punger and West (2013) have concluded that estimates of health impacts from exposure to ozone and premature mortality can be captured in models at these resolutions. We now perform a deeper analysis, however, of the response of atmospheric composition (rather than just health outcomes) to emissions in the model. In particular, the new section 3.3 uses a series of sensitivity analyses to quantify how ozone changes at different altitudes, the role of chemical lifetime compared to vertical transport, the degree to which this varies between seasons, how much of the air quality response is due to cruise altitude emissions versus landing and takeoff, the role of $NO_x$ compared to other aviation emissions, and (where possible) the sensitivity of these factors to model resolution.*

**Page 2, L56: Why so much lower? Can you summarize?**

*Based on the results from this study, we believe that the results from Vennam et al. (2017) reflected the fact that their nested simulations employed fixed boundary conditions. The use of fixed boundary conditions would mean that aircraft emissions being advected out of the domain could not influence surface conditions. Since our results show that US emissions cause increases in ozone which are hemispheric in nature, this oversight in Vennam et al. (2017) likely resulted in the loss of the key mechanism by which US aviation emissions could affect US surface air quality. This is highlighted on lines 501-506 of the main text.*

**Page 2, L62: Global total from aviation?**

*Correct – we now specify this explicitly in the text (line 65).*

**Page 3, L72: can you explain how you can equate climate impact with air quality impact? This seems not to be a scientific question.**

*We believe that this is a policy concern which requires scientific input. Decision makers such as the FAA and ICAO cannot perform monetization and prioritization of emissions control strategies (a political concern) without accurate quantification of both the climate and air quality impacts of aviation emissions (a scientific concern). This is of particular relevance now due to ongoing debate regarding whether to prioritize reduction in $NO_x$ emissions (motivate by air quality concerns) or improvement in fuel efficiency (motivated by climate concerns), with a key deciding factor being uncertainty regarding the true air quality impacts of $NO_x$. We cite the need for accurate quantification as part of the motivation of this paper but we do not claim to provide a mechanism for comparison between climate and air quality impacts.*

**Page 3, L79: All of these resolutions are using basically the same type of model set up. What would be the impact of using a more detailed treatment that would explicitly resolve the non-linear nature of exposure? Are there parameterizations for this from regional air quality models? That would be more interesting.**

*We agree that it would be interesting to also evaluate how and why regional air quality models find different results than global-scale models. However, one of our goals in this work is to understand why a single, consistent global model might find substantial differences in impact when only a single parameter – the grid resolution on which chemistry and transport are calculated – is changed. The fact that we find a 20% increase in estimated impacts as a result of this change alone suggests that it is in important factor in understanding disagreement between prior studies, and should be considered alongside factors such as differences in the parameterization of different physical processes between models. We now state this explicitly in the discussion on lines 495-497.*

**Page 3, L81: Please explain what the coded resolutions refer to the first time it is used.**

*We now do not introduce the "CN" resolution names until they can be properly explained.*

**Page 3, L94: Are the in and out emissions impacts linear? E.g., if you take Global - NoNA + NoNA - Off, do you get the same answer as Global - Off?**

*Since we only perform three simulations (with global aviation, with all aviation except that over the US, and without any aviation), we cannot directly evaluate the linearity of the response. This would require an additional simulation in which we only include aviation over the US (say, "NAOnly") – at which point we could evaluate the degree to which the effects of "NAOnly – Off" differ from the effects of "Global – NoNA".*

**Page 4, L98: why do you say CN for resolution and then use C?**

*The N in this case is intended to correspond to the number, hence the phrasing "Model resolutions are therefore denoted as CN where N is the number of grid elements along each edge of the cube" (lines 101-102).*

**Page 4, L104: What happens is you try to run at much higher resolution (say 5-10km). Can you do this, even for a limited time to gauge the physical impact on O3 and NOx? Using the same chemical mechanism. Versions of the MERRA-2 system have been run at these resolutions.**

*We did investigate the run time required to do this, as we agree it would be a valuable experiment. However, the combination of the high computational cost of the chemical integrator in GEOS-Chem and the need for at least a minimal spin-up period made this prohibitive. For context, a simulation at ~100 km resolution (C90) on 192 CPU cores takes 12 hours to complete one month. Even with perfect scalability and assuming no decrease in the model time step, a simulation at 10 km globally would require 100 times the resources (i.e. 230,400 CPU hours) to complete one month. The previous work with MERRA-2 at these resolutions was completed with an older version of GEOS-Chem using NASA resources, and required 3.5 million CPU hours to complete a year* (Hu et al. 2018). *We unfortunately do not have access to the necessary resources to consider such a simulation at this time.*

**Page 4, L119: Are the same MERRA2 data used and just averaged or binned to lower resolution? How high resolution does the MERRA-2 data go? What if you used one of the GEOS 'nature runs' at 3 or 7km?**

*We perform area-conservative regridding (averaging) over MERRA-2 data on each model level, preserving the full vertical resolution (72 levels) but reducing the horizontal resolution as necessary* (Eastham et al. 2018; Martin et al. 2022). *As discussed above we unfortunately do not have the computational resources to perform a simulation at 7 km resolution. However we have applied what resources we have available to produce the sensitivity simulations discussed above which we believe have helped to provide valuable additional insight into how and why aviation affects air quality, the degree to which resolution affects the simulated answer, and the role of in-domain versus out-of-domain emissions.*

**Page 6, L144: How large are the biases? If they are larger than the signal, then could the non-linearities impact the results?**

*It is hard to say whether biases in the base simulation could cause a systematic effect in the results, hence the caveat provided here. Since aviation emissions cause a relatively small change in surface concentrations, it is difficult to develop strong empirical constraints on which the specific ability of a model to simulate aviation-*

*attributable change in surface conditions can be evaluated. We now attempt to clarify this point in Section 3.4 while also providing additional context for some of the sources of uncertainty which could not be quantified.*

**Page 6, L153: Given that the model is uncertain and you are dealing with small perturbations, can you give an uncertainty estimate for (a) the difference in concentration due to aviation and (b) the uncertainty in your mortality estimates?**

*We now explicitly state in the abstract that any model-based estimate of the physical response of the atmosphere to aviation emissions is inherently uncertain (lines 28-30). However, as specified above we now include both an assessment of the degree to which coarse-resolution modeling introduces errors (end of Section 3.3) and limited exploration of potential sources of uncertainty (Section 3.4), alongside an expanded comparison of our results against those from other studies including those using other atmospheric models (Section 3.5).*

**Page 6, L165: You base the ozone values on one study? That doesn't seem appropriate. There must be others you can refer to?**

*We agree that more studies are needed to investigate the impacts of chronic exposure to ozone. We use Turner et al. (2015) as our core study as it remains the most recent epidemiological study of the health impacts of chronic exposure to ozone, was previously identified as a key study for understanding the potential global burden of disease resulting from ozone exposure (Malley et al. 2017; Seltzer et al. 2020) as well as aviation specifically (Quadros, Snellen, and Dedoussi 2020), and was itself based on a high-quality cohort (that from the American Cancer Society Cancer Prevention Study-II). The other major study of health impacts resulting from chronic exposure to ozone that we are aware of is Jerrett et al. (2009), which is based on the same cohort and appears to be superseded by Turner et al. (2015). This reflects the relative lack of epidemiological data for ozone exposure, which is of critical importance for this topic given that aviation appears to have a nearly unique influence on northern hemispheric ozone. Accordingly, we now state (lines 437-438) that "[g]iven that there are relatively few studies of the health impacts of chronic exposure to ozone compared to exposure to $PM_{2.5}$, this uncertainty is a key area of future research".*

**Page 7, L172: Here you describe the uncertainty in mortality rates: why is this not propagated through, especially to the abstract.**

*This was an oversight, and uncertainty bounds are now given in both the abstract and the discussion. We also explicitly state in the abstract that the uncertainty quantified is only due to uncertainty in the health response.*

**Page 7, L182: Again, these uncertainty estimates need to be in the abstract.**

*This is now the case.*

**Page 7, L185: The uncertainty interval for PM2.5 is ±8%? How different is that than a previous estimate? That seems ridiculously low. Also: what is the uncertainty due to the physical model difference in PM? This uncertainty is only due to epidemiology right?**

*We agree that this was misleading as it reflects only one potential source of uncertainty. We now explicitly state that this is the confidence interval due to uncertainties in the health response, and include a specific discussion (Section 3.4) of other sources of uncertainty in the atmospheric (rather than health) response to aviation emissions.*

**Page 7, L186: The ozone mortality estimated uncertainty is ±30%. Yet the difference from previous work is 560%! Is that entirely due to differences in the epidemiological assumptions?**

*Yes – as outlined above, we have endeavored to make clearer to the reader the difference between epidemiological and physical uncertainty.*

**Page 7, L191: Figure 2 is hard to interpret due to use of the default python color scale. Maybe get rid of anything which is not significant (make those points white, not a dominant blue): you need a definition of what is a significant difference and what is just noise. I suggest the variability either within a year, or between multiple years could give you a standard deviation here.**

*Due to the prohibitive computational cost of our simulations we only have one year of simulation results at C180, although we do now include discussion of (for example) the uncertainty associated with different meteorological years based on prior evaluations in the literature. However, since these results are generated using a chemistry*

*transport model, chaotic climate feedbacks are not present in our results. As such there is not a convenient definition of significance with regards to the change in concentrations.*

**Page 7, L191: Why would the O3 changes from aviation AT THE SURFACE be largest in the W. US and Tibetan Plateau? Does that really make sense? I suggest there is a transport problem with the model due to a terrain following coordinate and anomalous cross-isentropic transport to elevated topography. This is a common problem with transport schemes in global models.**

*This is a well-established result for aviation emissions, present in (for example) Barrett et al. (2010), Vennam et al. (2017), Phoenix et al. (2019) and most results presented in the multi-model intercomparison of Cameron et al. (2017). A reproduction of the result from the hemispheric-scale model used by Vennam et al. (2017) is given below, showing the elevated response over the Western US and Tibetan Plateau.*

[Figure]

*Figure 1. Surface ozone change calculated by Vennam et al. (2017) showing the same distinctive pattern of surface ozone change.*

*The effect of stratospheric ozone intrusions increasing surface ozone in the Western US has also been extensively discussed in the literature outside of the impact of aviation* (Lefohn et al. 2011; X. Wang et al. 2020; Lin et al. 2012)*, and the same is true for the Tibetan plateau* (Yang et al. 2022)*. We concur with those prior researchers that, rather than this being a pure model artifact, it reflects the fact that these locations are more strongly influenced by air from the upper troposphere and lower stratosphere which will in turn have been more strongly affected by aviation.*

*This is one of the interesting components of aviation's impacts which we believe makes it worthy of study, as the impact of the aviation sector on near-surface atmospheric composition is essentially unique compared to other industries.*

**Page 7, L193: So these particulates are secondary aerosols produced where there already is air pollution?**

*This is exactly correct.*

**Page 8, L205: This seems anomalous in the model. Does observed near surface air in the Western US have higher background O3 than elsewhere? Representative of transport of higher altitude air to the surface?**

*Precisely. A particularly good discussion of this is given in Lin et al.* (2012, 2015)*, where they show that surface ozone in the Western US is strongly affected by high altitude ozone, especially during the springtime. As a result, aviation's effect on surface ozone is also maximized in such locations and during the spring – because the high altitude air where aviation has the largest effect on ozone is being effectively carried to the surface. We now cite Lin et al. (2012) in the main text at this location (lines 236-237).*

**Page 8, L218: This begs the question: are the epidemiological effects linear? Does a 10 ng/m3 change in PM2.5 have the same impact if the background is 10ng/m3 (100% increase) or 100ng/m3 (10% increase)?**

*No, the epidemiological effects are not linear. For both ozone and PM$_{2.5}$ we use a non-linear concentration response function as detailed in Section 2.2. We therefore account for the baseline concentration in our calculation, as detailed in equation 1 of the main text. However the change in concentration associated with aviation is sufficiently small (of the order of 1% of baseline concentrations, as discussed by Lee et al. (2013) and Phoenix et al. (2019)) that there would not be expected to be a significant nonlinearity of the concentration response function between 0% and 100% of current aviation emissions.*

**Page 9, L220: Given the importance of secondary particulate matter, how uncertain is this production in the model? It seems as if this is one of the most uncertain elements.**

*We agree that the model's representation of secondary particulate matter is a key uncertainty and now highlight this in Section 3.4. On one hand, GEOS-Chem incorporates state-of-the-art modeling of secondary inorganic aerosol thermodynamics and partitioning, including nitrate aerosol, as represented by the ISORROPIA-II model (Fountoukis and Nenes 2007) with continuous updates and improvements from a community of dedicated researchers such as the implementation of nitrate photolysis (Shah et al. 2023), hydroxymethanesulfonate chemistry and its relationship to sulfate aerosol (Moch et al. 2020), and the entrainment and removal of NO$_y$ in clouds (Holmes et al. 2019). On the other hand this remains an area of active and ongoing research, and uncertainty in overall nitrate aerosol production in particular remains high. We now call this out specifically on lines 420-426.*

**Page 9, L238: are these absolute or relative values? If relative they are small, but if absolute they are large. And it's confusing given the the previous percentages if they are relative.**

*All differences are relative. We now state (line 266) that "All differences are given relative to the aviation-attributable change calculated at C180 unless otherwise stated".*

**Page 10, L252: see earlier comment: are these linear with the total effect over the US?**

*As discussed in response to the prior comment they are not; we account for differences in the baseline concentration from cell to cell.*

**Page 11, L260: Figure 5: similar color scale problem to figure 2: you should blank regions without significant changes.**

*Since we are using a chemistry transport model, we do not have a meaningful way to quantify significance (and do not have the resources to run additional years at the resolution shown here).*

**Page 12, L281: So you get exactly the same answer? That's suspicious. Are you using the same exposure and same modeling tools to do this? What is the difference in their ozone and PM2.5?**

*In this case we believe that it is a combination of a genuine result (more fuel burn is likely to mean more impact, after all) and coincidence. Quadros et al. (2020) do use a closely related modeling tool, but in their case they apply a nested approach rather than using a globally consistent resolution. They also use the same exposure assessment approach. There are certainly differences however, as demonstrated by the fact that our application of an updated version of the same model can result in a 15% change in estimated mortalities (explicitly called out in Section 3.4). Quadros et al. (2020) also do not find an identical split between ozone and PM$_{2.5}$, and this is now stated in Section 3.5 (line 431).*

**Page 12, L287: I think this needs to be in the main text.**

*Given the increased length of the article following these responses, we are reluctant to move more material to the main text. As such we have opted to keep the more detailed evaluation of sensitivity to CRF in the Supplemental Information for now.*

**Page 12, L299: how much is attributable to any physical differences in the change in PM2.5 and Ozone? Initial background state and model biases?**

*Since both models in this comparison are chemistry-transport models rather than climate models, we assume that the role of the initial background state is negligible. However we do now include a comparison of the physical response calculated in our model to that from a range of other studies (lines 441-450), showing that the estimated*

*change in surface-level ozone and PM₂.₅ is within the range reported from other studies (per TgN emitted). We also have a new section dedicated to the underlying physics and chemistry of aviation air quality impacts, including potential causes of the changes we observe between resolutions and between in-domain and out-of-domain emissions (Section 3.3).*

**Page 13, L322: PM2.5 emissions are not due to NOx are they? Maybe you need to remind the reader that NOx emissions are what change ozone since NOx is not mentioned in the results section.**

*We are grateful to the reviewer for highlighting the fact that we did not explicitly point out in the results section the role of aviation $NO_x$. We now include a brief recap of the fact that aviation $NO_x$ has been shown in previous studies to be the dominant contributor to changes in surface-level ozone and PM₂.₅ (Whitt et al. 2011; Prashanth et al. 2022; Eastham and Barrett 2016) on lines 329-332. We have also performed a short (6 month, covering the critical winter season) sensitivity simulation in which we include all aviation emissions except for $NO_x$. We find that the aviation-attributable wintertime increase in ozone and PM₂.₅ is decreased by 108 and 103% respectively if aviation $NO_x$ is not emitted. This is now stated on lines 373-375 of the main text.*

**Page 13, L328: I think this is disingenuous since you have really just limited the resolution differences to interpolation of input data. A proper assessment of resolution would alter the balance between parameterized and resolved quantities, but you do none of that and test only a limited range of resolutions and then apply them to a much higher resolution (1km) population data set. Which seems a big scale mismatch.**

*It is true that there is a large scale mismatch between the resolution at which these global assessments are performed and the resolution of the population dataset. We now include a statement in the discussion caveating the fact that our findings are specific to the question of how differences in resolution might affect global-scale assessments within a single model (lines 495-497).*

*Our objective is to help researchers and policy makers understand the different elements which cause differences between investigations of the same phenomenon, one of which is that different models are run at different resolutions. We also hope to help them to allocate resources, as our findings indicate that a researcher interested in understanding the global-scale impacts of aviation on surface air quality could run eight simulations at ~100 km for the cost of running one at ~50 km and still get results which match to within 2.5%.*

**Page 13, L332: Why would inconsistent models yield a smaller difference? Is that chance? I would expect a larger difference perhaps?**

*We realize that the emphasis in this paragraph incorrectly implied that inconsistency in model would inherently result in a smaller difference. We have rephrased the paragraph to instead emphasize that the causes of differences between our results and those in Yim et al. (2015) are different to the causes of differences with the results in Vennam et al. (2017). We also now emphasize that the differences found by Yim et al. (2015) which they associate with resolution could also be explained by model inconsistency, given the range of results reported by Cameron et al. (2017) in a model intercomparison. By contrast, the results of Vennam et al. (2017) are not within this range, and appear to be better explained by the use of a fixed boundary condition. We now explore this issue in more detail in the results on lines 360-372.*

**Page 14, L335: This is difficult to follow. Are you saying that Venam et al had non US aviation emissions in both? Wouldn't this tend to reduce changes 'neglecting the larger response'? So why is the response larger? I don't follow the logic.**

*This paragraph has been rewritten as described above.*

**Page 14, L351: You lost your uncertainty range again. Please put it here and in the abstract.**

*We apologize for this oversight. Uncertainty ranges are now included in the conclusions and abstract.*

**Page 14, L352: How much uncertainty results from the different meteorology in 2005 v. 2015?**

*We now cite previous estimates of the effect of meteorology on exposure to ozone and fine particulate matter resulting from aviation emissions.*

*Referee #5*

**This manuscript presented a modeling analysis of global health impact associated with ozone exposure due to aviation emission. Impacts of anthropogenic emissions on air pollution have been thoroughly investigated over inland areas, yet the contribution from aviation remains poorly documented. The study applied a solid modelling tool and focused on an interesting topic. But there are two critical main issues need to be addressed before the acceptance could be considered.**

*We appreciate the constructive feedback and have endeavored to address the two critical issues raised by the reviewer.*

**First, the manuscript proposed a very interesting question at the introduction section but didn't mention it in the discussion or conclusion. Line #57 mentioned: "whether the air quality impacts of aviation are dominated by local sources or are the result of larger atmospheric changes.", and line#63 mentioned "there questions urgently need to be resolved". So the introduction indicated this is one of the question that would be at least discussed in this study but unfortunately no such discussion was mentioned later. Section3.2 provided a detailed comparison of the estimated mortalities between this study and previous studies, but the most significant difference was due to using a more epidemiological data, while the proposed question remain unsolved.**

*We agree that the original manuscript did not spend enough time addressing this central question – of the degree to which local versus global aviation influences surface air quality. We have separated out and expanded a new section of the results dedicated to this question (Section 3.2, "In-region versus out-of-region emissions"). This section first focuses the reader on the results which show that emissions from US aviation contribute roughly proportionally to surface-level ozone across the Northern Hemisphere, rather than having an effect which is mostly isolated to the region of emission. We now also include a discussion exploring why this is the case, which focuses on the simulations in which aviation emissions over the US are excluded. By subtracting those results from the simulation in which all aviation emissions are included, we observe how atmospheric composition is affected by US aviation emissions only. For the purposes of this response we have performed some additional analysis (below); although this is not included in the main text, it informed our discussion on lines 360-372 of why in-region emissions do not dominate aviation emissions impacts.*

*The change in mean ozone between 15 and 45°N for local winter and summer shows two key features (*Figure R 4*). First, we find that the overall cruise-altitude increase in ozone over the US is actually greater during summer than during winter. However, we also find that it is more localized in both longitude and altitude.*

[Figure]

*Figure R 4. Changes in mean ozone between 15 and 45°N (area-weighted) during northern-hemispheric winter (DJF, left) and summer (JJA, right) due to aviation over the US. Results are from the sensitivity simulations conducted at C90, for which additional data were archived.*

*As discussed in the response to Referee #4, this is likely due to the wintertime combination of longer lifetimes with faster vertical transport from cruise altitudes to the surface. The net result is that aviation has a nearly unique signal. The regional influence decreases by 38% from winter to summer, such that the effect of US aviation on US ozone decreases from +0.35 ppbv in the winter to +0.22 ppbv in the summer. Meanwhile, the effect of US aviation on the rest of the world is decreased by 79% over the same period, with the population-weighted mean exposure to ozone falling from +0.27 ppbv in winter to +0.056 ppbv in summer (*Figure R 5*).*

[Figure]

*Figure R 5. Change in surface-level ozone worldwide due to US aviation in winter (DJF, left) and summer (JJA, right). Results are here from the core simulation at C180.*

*This effect is exceeded by the greater overall effect of aviation on surface level ozone during winter, when the impact is more diffuse. When all aviation is simulated, surface level ozone over the US is increased by 1.6 ppbv during winter compared to 0.64 ppbv during summer (a 60% reduction). This implies that US aviation is responsible for 21% of aviation-attributable exposure in the US during winter, and 34% during summer. However, since the overall summertime exposure is lower, the wintertime effect – where global aviation is responsible for 79% of the surface ozone change in the US - is dominant.*

*For PM$_{2.5}$, we find a similar outcome with a larger in-region signal. During summertime, US aviation is responsible for an additional 21 ng/m$^3$ of US PM$_{2.5}$, 55% of the total effect from global aviation. During wintertime, the absolute contribution of US emissions increases to 49 ng/m$^3$, but the relative contribution falls to 31%. As in the case of aviation-attributable ozone, in-region emissions have a larger relative effect during summer, but the largest absolute contribution occurs during winter and is the result of out-of-region emissions.*

*This finding has significant implications for policy makers. Regardless of the season, out-of-region emissions are the dominant contributor to in-region impacts, even for a region with as much fuel burn and covering as much physical space as the US. Similarly, exposure to both PM$_{2.5}$ and ozone are greater during winter, when the contribution of in-region emissions is reduced. However, in-region emissions have a greater influence on PM$_{2.5}$ than on ozone, and (due to the fact that baseline summertime ozone exposures are typically greater) few studies have considered whether an increase in wintertime ozone might have less severe health consequences than an increase in summertime ozone. If indeed seasonal changes in ozone exposure are found to be important, then a focus on regional aviation emissions may still have disproportionate benefits for public health.*

**Second, as a modeling study the manuscript lacks a necessary evaluation section. Almost all discussions were made based on simulation results, so it is very important to demonstrate the reliability or remaining uncertainty of the simulation results. Without solid evaluation the rest of discussion regarding contributions of aviation to air pollution and associated mortalities would be difficult to interpret.**

*GEOS-Chem has been extensively evaluated for its ability to reproduce global observations of both atmospheric composition and air quality, and has also been widely used in air quality assessment. However, we recognize that these results still reflect only those of a single model. We therefore provide both citations of recent assessments and a statement to this effect on lines 143-150. We have also endeavored to improve and extend our comparison to include results from models other than GEOS-Chem (see Section 3.5).*

**There are a few other minor issues with the organization of the manuscript. For example, line #80 mentioned simulations were conducted for 400, 100, and 50km and section 3 demonstrated the differences between. Apparently finer resolution can better reproduce atmospheric chemistry especially for ozone which is sensitive to mixing of NOx and VOCs. The manuscript indeed shows a large difference between fine and coarse simulations, but this seems more like a technical improvement other than a key innovative science finding.**

*We respectfully disagree with the assessment that this is purely technical, but recognize that the manuscript as it was previously organized did not sufficiently highlight the scientific findings. We hope that the changes made in response to the comments above help in this respect, but we have also worked to try and more prominently highlight the following findings:*

- *First, that aviation's impacts on air quality are global in nature and not dominated by near-airport emissions. This is now more specifically illustrated through our analysis of the relative role of near-airport and regional-scale air quality (lines 283-292), and our quantification of the relative role of cruise-altitude $NO_x$ emissions (as opposed to non-$NO_x$ emissions (line 375) and LTO emissions (line 376)).*
- *Second, the finding that in-region emissions of $NO_x$ from aircraft do not cause the majority of air quality impacts associated with aviation (Section 3.2), and that this is largely the result of seasonal differences in ozone chemistry and transport (Section 3.3).*
- *Third, that the effect of aviation emissions on global air quality is greater than previously estimated and dominated by ozone – making it almost unique in comparison to other sectors (opening of Section 3).*
- *Finally, the finding that moderate-resolution simulations (~100 km) are sufficient to capture impacts that are resolved at 50 km, but that low resolution (~400 km) simulations are likely to underestimate impacts (Section 3.1).*

*In addition to the above points, we believe that this manuscript helps to address a standing point of confusion in the literature – that being whether very high resolution simulations would be expected to significantly reduce the simulated impact of aviation. This finding was reported by Vennam et al. (2017) and was a possible explanation for the results of Yim et al. (2015). Our work provides a plausible explanation for both outcomes, while also demonstrating that the hemisphere-scale increase in surface-level ozone and $PM_{2.5}$ associated with cruise-altitude aviation is unlikely to disappear as model resolution continues to advance.*

Thank you for considering our submission for *Atmospheric Chemistry and Physics*. We hope you agree that these comments have been addressed and look forward to your response.

Regards,

Sebastian Eastham

[revised manuscript text omitted]

---

## Author Response (AR2)

Massachusetts Institute of Technology 77 Mass. Ave. Office E19-439F, Cambridge MA 02139, USA http://globalchange.mit.edu | http://lae.mit.edu seastham@mit.edu | (617) 253-2170

Editorial Office

Atmospheric Chemistry and Physics

January 10th, 2024

Dear Dr. Querol and reviewers,

**Second revised submission of "Global impacts of aviation on air quality evaluated at high resolution" to *Atmospheric Chemistry and Physics**

Thank you for your time and effort in considering our manuscript a second time. We greatly appreciate the comments made by the reviewers, and have again made an effort to address their concerns. A response to each reviewer comment is given below; the original comments are shown in **bold** with our responses shown in *italics*.

**Referee #1**

The revised manuscript has been greatly improved, it provided a more in-depth discussion of why and how aviation affected air pollution and a evaluation of the mortality impact. My previous comments were also properly addressed. Therefore, I would recommend it to be accepted for publish.

We thank the reviewer for their comments and their encouragement.

**Referee #2**

I appreciate the authors responding to the detailed reviews and comments of myself and the other reviewers. I think they have done a decent job in this regard.

We thank the reviewer for their assessment.

However, the two main points I still think remain.

1. I dislike the use of mortality and having the main uncertainty derived from public heath studies. The authors indicate ACP is publishing this sort of work, so that may justify it in the editor's eyes, but I don't think it's really appropriate. Maybe it's just my skepticism of getting big numbers of deaths from very tiny changes. So discount that if you want. I'm not a public health expert (but also this is not Atmospheric Chemistry or Physics).

We appreciate that this is a philosophical question. However, based on review of other similar publications in ACP, in addition to an increased focus in our manuscript on both the question of how aviation affects air quality (Section 3.3) and how the specific approach to atmospheric modeling affects the simulated outcomes (Sections 3.2 and 3.4), we believe that our work is appropriate to this journal.

2. The resolution issue is a bit weak, and by doing more low resolution runs, they have not helped their cause here. I don't think the paper adds much except to say resolution doesn't matter at the scales they are looking at, all else being equal. Since there is no downscaling I am not surprised.

We respectfully disagree with this assessment. Firstly, we do find a substantial resolution effect when comparing coarse (400 km) and higher-resolution (100 or 50 km) simulations. As stated in the abstract, simulations at coarse resolution (~400 km) underestimate the air quality impacts of aviation by ~17% compared to simulations at resolutions of 100 or 50 km (i.e. the use of high-resolution modeling increases the estimated impact by 20%). This means that almost all prior studies of aviation's impacts – including those performed in the intercomparitive study

by Cameron et al. (2017) which relied exclusively on models at resolutions of ~200 and 400 km – included a thenunknown underestimate.

Furthermore, the finding that a resolution change from 100 to 50 km results in little change is in contravention to an existing study. As detailed in Section 3.5 of the manuscript, Vennam et al. (2017) found an order of magnitude decrease in air quality impacts associated with aviation when using a higher-resolution (~40 km) nested model than when using a lower-resolution (~100 km) hemispheric model. This result was recently used as the basis for a study by Lee et al. (2023) to justify neglecting aviation's effects on air quality in favor of focusing only on aviation  $CO_2$ . Our work specifically refutes this by showing that, when a single, consistent, global model simulates aviation's effects at similar resolutions (~100 and ~50 km), we instead find that there is no such decrease in impact. To help make this point more clearly in the manuscript we now make it explicitly on lines 59-61 and 511-513 of the manuscript.

**Referee #5**

This study utilizes the GEOS-Chem model to perform simulation experiments to evaluate the impact of aviation emissions on air quality and associated health effects. The authors also explore the influences of both local sources and transported pollutants. Overall, the manuscript is well-written, and the manuscript quality has notably improved based on suggestions from reviewers. I believe that the content related to air quality and its health effects aligns with the scope of ACP. Consequently, I recommend the publication of this study. However, before publication, the authors may consider the following points outlined below:

We thank the reviewer for their assessment and review.

**Comments**

1. It would be better to provide a Table to describe the configurations of different simulations.

We agree and now do so in Table S1, which is now referred to on line 218.

2. The authors put a lot of effort into the simulations. However, it seems that the results of different simulations are not sufficiently demonstrated. For example, the influence of different model resolutions is only exhibited in Figure 4. It would be more helpful if the authors could produce figures (similar to Figures 5-7) with other resolutions to demonstrate the possible impact of different model resolutions on horizontal and vertical distributions. Similarly, it would be more interesting if the authors could provide figures (similar to Figure 5) to show the contributions of various sources (such as US-domain emissions) and transport more clearly.

We do now include a graphical comparison of the effect of model resolution in Figure S2. Although we chose to exclude this from the main text due to the length of the manuscript, we do discuss this figure on lines 288-296. We would be happy to bring this figure into the main text, however, if it is considered to significantly improve the manuscript.

Unfortunately the simulations which provided ozone production and loss rates were only performed at one resolution, meaning that we cannot extend Figure 5 to the coarser and finer resolutions. However, we agree that it is interesting to see how the different resolutions affect not only the horizontal but also the vertical distribution of aviation-attributable ozone. We therefore now include a new evaluation (Figure S3, reproduced below) and analysis of the differences in zonal mean ozone throughout the Northern Hemisphere for the simulations at 50, 100, and 400 km (lines 400-403).

Figure R 1. Annual mean aviation-attributable change in zonal mean ozone at C180 (top), C90 (middle), and C24 (bottom) for the Northern Hemisphere. Contours show the difference between the estimate at that resolution and at C180, with positive values indicating that the simulation at the lower resolution is overestimating the change in ozone relative to the simulation at C180. Each contour corresponds to a 1.5 ppbv difference. This is reproduced as Figure S3 in the Supplemental Information.

**3. Figure 4: the contribution of US-domain aviation emissions to surface PM2.5 concentrations in China is larger than that in the US itself. It is hard to understand and needs to be explained.**

This is due to the higher concentration of particulate matter precursor gases over China than over the US, meaning that the same quantity of aviation-attributable oxidant would result in more additional particulate matter. We now explain and quantify this effect on lines 318-321.

Thank you for considering our submission for *Atmospheric Chemistry and Physics*. We hope you agree that the comments and concerns raised have been addressed and look forward to your response.

Regards,

Sebastian Eastham

**References**

- Cameron, M. A., M. Z. Jacobson, S. R. H. Barrett, H. Bian, C. C. Chen, S. D. Eastham, A. Gettelman, et al. 2017.
  "An Intercomparative Study of the Effects of Aircraft Emissions on Surface Air Quality." *Journal of Geophysical Research, D: Atmospheres* 122 (15): 8325–44.
- Lee, David S., Myles R. Allen, Nicholas Cumpsty, Bethan Owen, Keith P. Shine, and Agnieszka Skowron. 2023. "Uncertainties in Mitigating Aviation Non-CO2 Emissions for Climate and Air Quality Using Hydrocarbon Fuels." *Environmental Science: Atmospheres*, November. https://doi.org/10.1039/D3EA00091E.
- Vennam, L. P., W. Vizuete, K. Talgo, M. Omary, F. S. Binkowski, J. Xing, R. Mathur, and S. Arunachalam. 2017.
  "Modeled Full-Flight Aircraft Emissions Impacts on Air Quality and Their Sensitivity to Grid Resolution." Journal of Geophysical Research, D: Atmospheres 122 (24): 13472–94.